# Inflammation triggers immediate rather than progressive changes in monocyte differentiation in the small intestine

Girmay Desalegn [1,2] & Oliver Pabst[1]

Bone marrow-derived circulating monocytes contribute to the replenishment and maintenance of the intestinal macrophage population. Intestinal monocytes undergo context-dependent phenotypic and functional adaptations to either maintain local immune balance or support intestinal inflammation. Here we use monocyte adoptive transfer to dissect the dynamics of monocyte-to-macrophage differentiation in normal and inflamed small intestine. We find that during homeostasis CCR2 and β7-integrin mediate constitutive homing of monocytes to the gut. By contrast, intestinal inflammation increases monocyte recruitment via CCR2, but not β7-integrin. In the non-inflamed intestine, monocytes gradually differentiate to express genes typically associated with tolerogenic macrophage functions. Conversely, immediately upon entry into the inflamed intestine, monocytes adapt a different expression pattern in a partly Trem-1-dependent manner. Our observations suggest that inflammation fundamentally changes the kinetics and modalities of monocyte differentiation in tissues.

[1] Institute of Molecular Medicine, RWTH Aachen University, D-52074 Aachen, Germany. [2] Institute of Immunology, Hannover Medical School, D-30625 Hannover, Germany. Correspondence and requests for materials should be addressed to O.P. (email: opabst@ukaachen.de)

Macrophages are mononuclear phagocytes (MNPs) that are found in all tissues of the body. In the gut, macrophages reside in all layers of the small intestine and colon; the lamina propria, muscularis externa, and serosa. Macrophages in the muscularis and serosa interact with enteric neurons to control intestinal motility but also provide tissue protection during injury[1]. The largest population of intestinal macrophages is present in the lamina propria, the tissue located directly beneath the gut epithelium. Thus, lamina propria macrophages are constantly confronted with luminal content and function as first-line sentinel immune cells for the host. Macrophages can also promote epithelial barrier integrity[2], help the repair and remodeling of damaged tissues[3], and can interact with T cells locally in the intestine[4,5]. Compared to macrophages in many other tissues, intestinal lamina propria macrophages are hyporesponsive to innate stimuli while maintaining their phagocytic and bactericidal properties[6–8]. These features allow macrophages to 'silently' clear bacteria and debris without triggering pathogenic inflammatory responses[8–10]. On the other hand, during intestinal inflammation, macrophages become a major source of proinflammatory mediators that further exacerbate inflammation[10–13]. This shows that intestinal macrophages display contrasting functions in normal and inflammation, and react in a highly context-dependent manner.

The contrasting functions of macrophages might be linked to their cellular origin and/or mostly reflect tissue- and context-dependent modulation. In fact, our understanding of the complex origin of tissue macrophages is still emerging. For more than four decades, tissue-resident macrophages were considered to derive from adult bone marrow (BM) monocytes[14]. More recently, fate mapping studies and experiments in parabiotic mice demonstrated that macrophages in most tissues of adult mice are of embryonic origin, including microglia in the brain[15,16], alveolar macrophages in the lung[16,17], Kupffer cells in liver and Langerhans cells in epidermis[16]. These macrophages originate from embryonic precursors which seed these tissues during development and are maintained by local self-renewal[16,18]. Like most other tissues, the gut becomes seeded during development by embryonic precursors and a fraction of those is maintained to adulthood[19]. Nonetheless, a substantial fraction of intestinal macrophages derive from circulating adult monocytes and are constantly replenished during adulthood[20].

Defining macrophage populations in the gut using reliable markers is still challenging. CX3CR1 expression, mostly through the use of a CX3CR1-GFP reporter mouse strain[21], has widely been used to define gut macrophages and analyze their function. An important refinement of identifying gut macrophages by flow cytometry came with the use of CD64 as an unambiguous marker discriminating macrophages and dendritic cells (DCs) in the gut[22] and recently Tim4 has been suggested as an attractive tool to discriminate between embryonic- and adult monocyte-derived gut macrophages[19]. Surface expression of Ly6C and MHCII along with CX3CR1 are commonly used to characterize monocyte differentiation by flow cytometry. Monocyte differentiation has been suggested to progress through distinct phenotypic developmental stages referred to as the ''monocyte waterfall'': incoming monocytes are Ly6C$^+$MHCII$^-$ cells differentiating via Ly6C$^+$ MHCII$^+$ intermediates into Ly6C$^-$MHCII$^+$ macrophages that eventually gain high expression of CX3CR1, characteristic of the macrophages in the normal adult gut lamina propria[11,22].

Different subsets of monocytes/macrophages are present during inflammation as compared to the non-inflamed state. In particular, a marked increase in Ly6C$^-$MHCII$^+$ cells with intermediate expression of CX3CR1 has been reported in the inflamed colon[11,12]. Since a phenotypically similar CX3CR1$^{int}$ population is also present in normal intestine, increase of this phenotype during inflammation has been suggested to result from an arrest of the monocyte-to-macrophage differentiation program[11,22]. In contrast to macrophages abundantly present in homeostasis, the CX3CR1$^{int}$ population abundantly produces inflammatory mediators such as TNF, IL-1, IL-6, and CCL2[10–13]. These proinflammatory monocytes/macrophages derive from the influx of circulating monocytes and exacerbate the inflammation. Both CCR2 and β7-integrin deficiency have been reported to ameliorate experimental colitis by reducing the monocyte trafficking to the inflamed colon[23–25]. Consistently, ablation of circulating monocytes by anti-CCR2 antibodies prevented severe colitis[12]. This suggests that the magnitude of monocyte recruitment and their subsequent local differentiation in the tissue are key determinants regulating homeostasis and inflammation in the gut. Yet, the mechanisms that allow for such context-dependent outcomes have not been fully defined.

Several complementary approaches have been used to analyse monocyte-to-macrophage differentiation. Fate tracking and more recently single-cell sequencing are powerful tools to define precursor-product relationships and deduce cell differentiation paths. Here we use an adoptive monocyte transfer system to track the fate of BM-derived circulating monocytes and investigate in-depth their development in the inflamed and non-inflamed small intestine using multi-parameter flow cytometric and transcriptional analysis. We show that monocytes entering non-inflamed and inflamed intestine undergo a fundamentally distinct differentiation program, suggesting that local environmental cues regulate monocyte plasticity in a context-dependent manner.

## Results

**Small intestinal inflammation increases monocyte recruitment.** Endogenous intestinal MNPs represent a mixture of cells at different developmental stages[11,19,26]. Thus, to fully dissect temporal dynamics and factors influencing monocyte differentiation, we used an adoptive transfer system approach to generate a synchronized population of monocytes whose development and differentiation can be tracked in vivo, over time.

BM cells were isolated from wild-type (WT) mice and CD11b$^+$ Ly6C$^+$ monocytes were purified by negative magnetic enrichment resulting in a 95% purity (Fig. 1a). Purified monocytes represented a homogenous population expressing CD11b and CX3CR1 but no MHCII (Fig. 1a, b). Purified monocytes were then intravenously transferred into non-manipulated recipients (Fig. 1c). One day after transfer, lamina propria cells were isolated from small intestine and adoptively transferred cells were identified by the expression of congenic markers. Consistent with observations obtained from colonic lamina propria[11,22], we observed a small but notable population of adoptively transferred cells in the small intestine of CCR2-deficient recipient mice (Fig. 1d). Recovery of adoptively transferred cells from the small intestine of WT recipients was lower as compared to CCR2-deficient recipients (Supplementary Fig. 2e), possibly reflecting differences in available niches or a competitive advantage of WT monocytes compared to endogenous CCR2-deficient cells. Thus, for further adoptive transfer experiments we used CCR2-deficient mice as recipients unless stated otherwise.

We next sought to characterize monocyte development in the normal and inflamed small intestine. Unlike the well-established different models of experimental colitis, only few mouse model of small intestinal sterile inflammation have been reported. In this study, we adopted a model of small intestinal inflammation triggered by surgical manipulation of the gut[27–29]. Consistent with Farro et al[29]., intestinal manipulation-induced inflammation in both WT and CCR2$^{-/-}$ mice. In contrast, laparotomy alone without mechanical manipulation of the gut (sham surgery) did

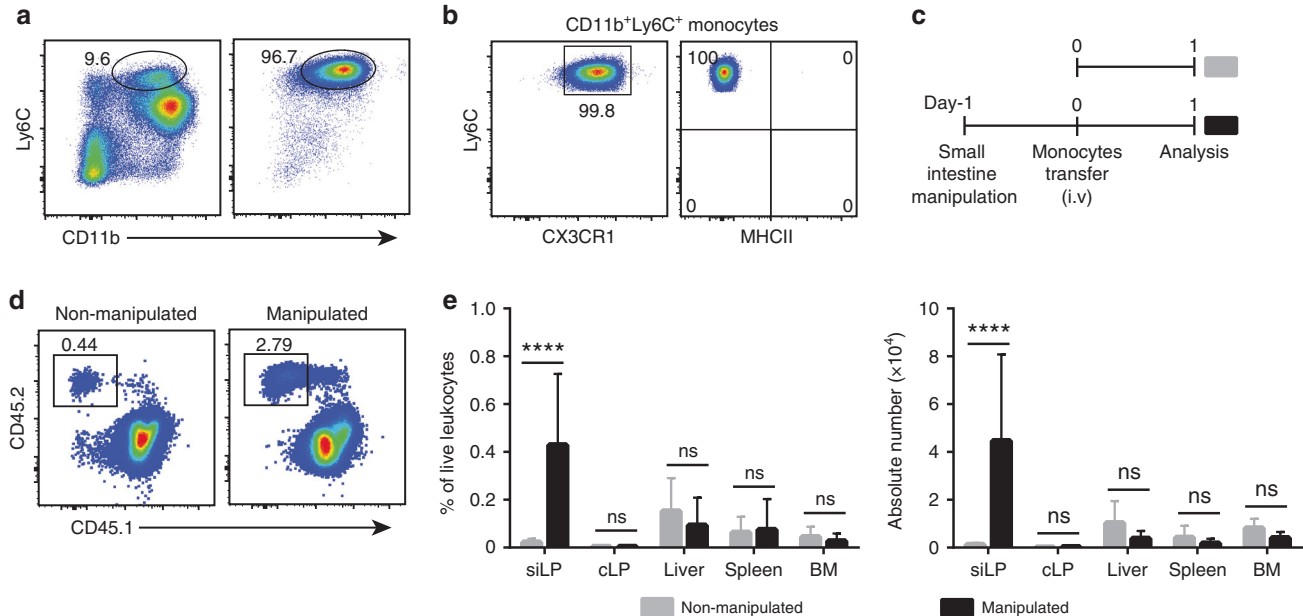

**Fig. 1** Small intestinal inflammation increases the recruitment of adoptively transferred monocytes. **a** Representative FACS plots demonstrating frequencies of CD11b⁺Ly6C⁺ monocytes among live leukocytes in bone marrow (BM) of CX3CR1$^{gfp/+}$ mice before (left panel) and after enrichment (right panel). **b** Expression of CX3CR1/GFP and MHCII on purified monocytes as indicated in **a**. **c** Schematic illustration of the experimental set-up for **d**, **e**. Monocytes were transferred into untreated recipients (non-manipulated mice) or one day after small intestinal manipulation (manipulated mice). **d** Frequencies of adoptively transferred CD45.2⁺ donor cells in small intestinal lamina propria (siLP) one day after transfer into non-manipulated and manipulated CD45.1⁺ CCR2-deficient recipients. FACS plots are gated on live CD11b⁺ cells. **e** Frequencies and total numbers of CD64⁺CD11b⁺Ly6C⁺ donor monocytes recovered from siLP, colonic lamina propria (cLP), liver, spleen, and BM. Bar charts illustrate data pooled from at least three independent experiments with a total of at least five mice each. Unpaired *t*-test; mean ± s.d.; ns not significant; ****$p < 0.0001$

not induce gut inflammation (Supplementary Fig. 1). Inflammation was associated with increased monocyte and neutrophil recruitment as well as reduced villus length and disrupted crypt architecture (Supplementary Figs. 1a, b and 2a-c).

Phenotypically, the endogenous MNP in manipulated CCR2-deficient mice closely reflected the populations observed in control mice (Supplementary Fig. 3). Hence, in this study we used CCR2-deficient mice that underwent small intestine manipulation one day before monocyte transfer as a model of small intestinal inflammation (Fig. 1c). To compare the efficiency of monocyte recruitment between non-manipulated and manipulated recipients, we systematically quantified the number of donor monocytes in small intestinal and colonic lamina propria, liver, spleen, and BM one day after transfer. Both frequency and total numbers of donor monocytes were significantly higher in the small intestine of manipulated as compared to non-manipulated recipients (Fig. 1d, e). Along with the recruitment of adoptively transferred monocytes, we noted an increased accumulation of endogenous Ly6C⁺ monocytes in manipulated small intestine (Supplementary Fig. 2d and 3d). Notably, migration of transferred monocytes into other tissues was unaffected by small intestinal manipulation. The frequency and absolute number of donor monocytes in the colon, liver, spleen, and BM were similar in manipulated and non-manipulated recipients (Fig. 1e). This shows that the inflammation triggered by surgical manipulation of the small intestine was largely restricted to the local environment with only limited systemic effects. Thus, the combination of adoptive monocyte transfer and small intestinal manipulation provides a powerful tool to compare the recruitment and differentiation of monocytes in normal and inflamed small intestine without the confounding factors of heterogeneity among the endogenous cells or major systemic inflammation.

**CCR2 but not β7-integrin mediates homing to inflamed gut.** Different immune cells, including lymphocytes, macrophages, and dendritic cells, constitutively enter the small intestine to sustain local populations. Lymphocyte migration into the small intestine generally relies on β7-integrin and the chemokine receptor CCR9[30,31], but it is not known if the same applies to monocytes. Experiments using mixed WT:CCR2$^{-/-}$ BM chimeras showed a role for CCR2 in the accumulation of monocytes in the small intestine and colon[20,22]. In fact, BM monocytes express CCR2, α4β7-integrin and low but detectable levels of CCR9 (Fig. 2a and Supplementary Fig. 4), suggesting that these factors might indeed be involved in monocyte recruitment to the gut. To systematically analyze the role of CCR2, β7-integrin, and CCR9 in recruiting monocytes from the circulation into non-inflamed and inflamed intestine, we performed competitive adoptive monocyte transfers. Monocytes were isolated from BM of WT or respective gene-deficient mice, differentially labelled and intravenously injected at a 1:1 ratio one day before analysis (Fig. 2b, c).

CCR2, β7-integrin, and CCR9 did not affect homing of the donor monocytes into spleen and liver of healthy/non-manipulated recipients as indicated by unchanged ratios of gene-deficient to WT monocytes in these organs (Fig. 2d). Similarly, WT and all gene-deficient monocytes efficiently entered the BM. CCR2-deficient monocytes, as compared to WT monocytes, were significantly enriched in the recipients' BM, possibly reflecting the known function of CCR2 in egress of monocytes from the BM[32,33]. In contrast, deficiency in CCR2 and β7-integrin significantly impaired accumulation of transferred monocytes in the small intestinal and colonic lamina propria of non-manipulated mice. β7-integrin-deficient monocytes were about 2.7-fold reduced in proportion in small intestine whereas CCR2-

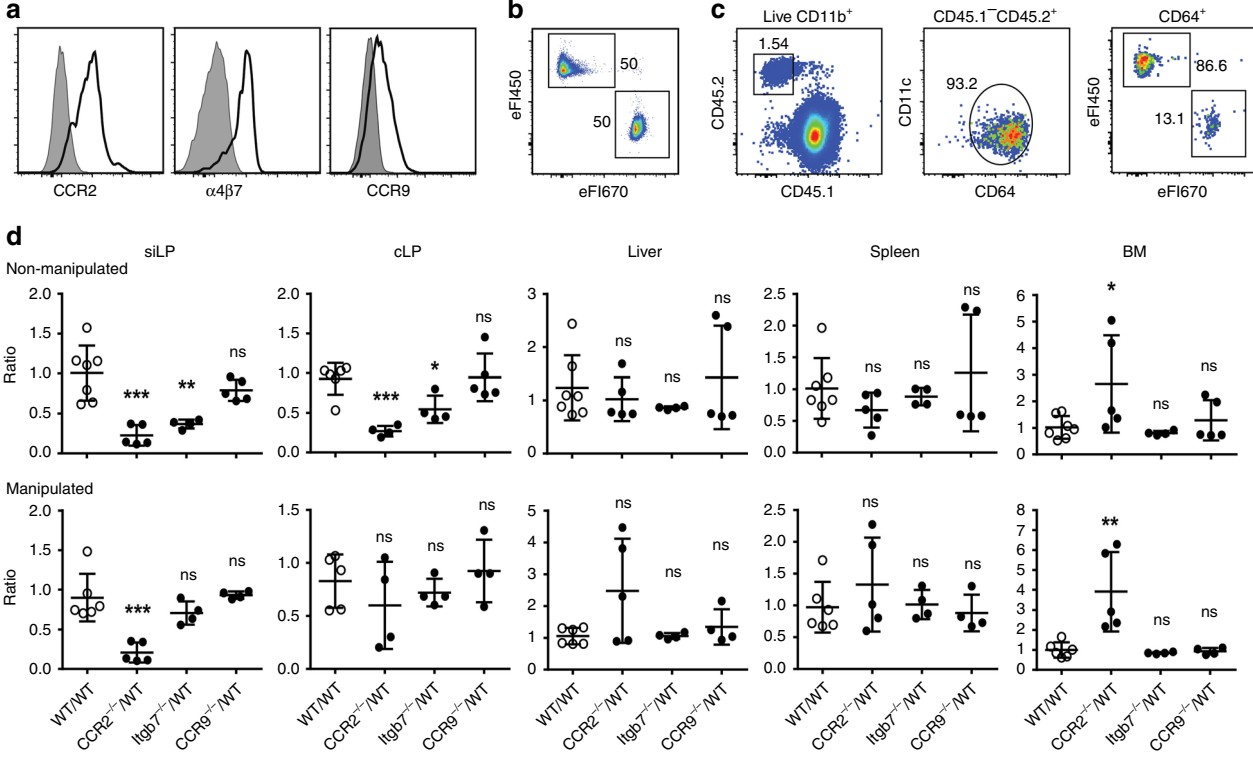

**Fig. 2** CCR2, but not β7-integrin, mediates homing of circulating monocytes to the inflamed intestine. **a** Purified CD11b⁺Ly6C⁺ BM monocytes express CCR2, α4β7-integrin, and CCR9. Isotype controls are shown as gray shaded histogram. **b** Monocytes were purified from BM of CD45.2⁺ WT and gene-deficient donors (CCR2, β7-integrin, or CCR9), differentially labeled with either eFluor 450 or eFluor 670 and mixed at a 1:1 ratio. **c** Mixtures of labeled monocytes were transferred into non-manipulated and manipulated CD45.1⁺ CCR2⁻/⁻ recipients. Transferred cells were identified by congenic markers (left panel) and the donor monocytes identified as CD11b⁺CD11c^low/int CD64⁺ cells (middle panel). The frequency of the differentially labeled donor monocytes (right panel) was analyzed one day after cell transfer to determine the ratio of gene-deficient to WT cells. **d** In small intestinal lamina propria (siLP), colonic lamina propria (cLP), liver, spleen, and BM, the ratio of gene-deficient to WT monocytes was determined 1 day after cell transfer as indicated in **c**. Data are displayed as relative change with respect to control experiments comparing differentially labeled WT monocytes. Data were pooled from two or more independent experiments. Symbols depict individual mice, *n* = 4–7 mice. One-way ANOVA followed by Sidak's multiple comparison tests; mean ± s.d.; ns not significant; *$p < 0.05$, **$p < 0.01$, ***$p < 0.001$

deficient monocytes showed an even stronger effect and were about 4.5-fold less frequent as compared to their WT counterparts (Fig. 2d). Similar sized effects were also observed in the colonic lamina propria. Despite its effects on lymphocytes, CCR9 deficiency did not affect accumulation of monocytes in either small intestine or colon as compared to WT cells. Extending the experiments to inflammatory conditions, equal mixtures of WT and gene-deficient monocytes were transferred to recipients that underwent small intestinal manipulation one day before transfer. Similar to the healthy situation, no differences between WT, CCR2, β7-integrin, and CCR9-deficient monocytes were observed in liver, spleen, and BM (with the exception of increased proportions of CCR2-deficient cells in BM already noted during homeostasis) (Fig. 2d). Interestingly, β7-integrin-deficient monocytes showed no competitive disadvantage in homing to the inflamed small intestine while the effect of CCR2 was preserved during inflammation (Fig. 2d). This could indicate that inflammation may significantly alter the molecular machinery orchestrating recruitment of monocytes.

**Monocytes gradually differentiate into CX3CR1^hi macrophages**. As monocytes enter tissues from the circulation, they are thought to gradually change their phenotype and function in a highly tissue-specific manner. Among live CD11b⁺Ly6G⁻SiglecF⁻ endogenous leukocytes in the small intestine, CD11c^lo/int CD64⁺

cells were defined as a bona fide monocytes and macrophages, excluding intestinal DCs (Fig. 3a). In this study, we will hereafter refer to this population as MNP^CD64⁺ population. This cell population can be split into three distinct subpopulations in the small intestine based on expression of Ly6C and MHCII: Ly6C⁺ MHCII⁻, Ly6C⁺MHCII⁺, and Ly6C⁻MHCII⁺ cells (Fig. 3a; Supplementary Fig. 3d, e;[22]). Ly6C⁻MHCII⁺ cells can be further divided based on expression of CX3CR1 into CX3CR1^int and CX3CR1^hi cells, the latter constituting the major population of intestinal MNPs (Fig. 3a; Supplementary Fig. 3e). It was suggested that these phenotypic subpopulations reflect different stages of monocyte differentiation in the small intestine and colon[11,22,34]. Consistent with this, we observed that, besides CX3CR1, surface expression of CD64, CD11c, and CD206 gradually increased as cells progressed from Ly6C⁺MHCII⁻ monocytes to Ly6C⁻ MHCII⁺CX3CR1^hi macrophages in the small intestine (Fig. 3b).

To confirm that the proposed model reflected the small intestinal MNP differentiation pathway, we analyzed the phenotype of adoptively transferred monocytes 1, 2, or 5 days after transfer into non-manipulated recipients (Fig. 3c). Congenic marker staining allowed the identification of a small but notable population of donor CD11c^lo/int CD64⁺ (MNP^CD64⁺) monocyte-derived cells (Fig. 3d). One day after transfer, on average 70% of the transferred monocytes were Ly6C⁺MHCII⁻, whereas the rest showed upregulation of MHCII (Fig. 3e, f). By day 2, most

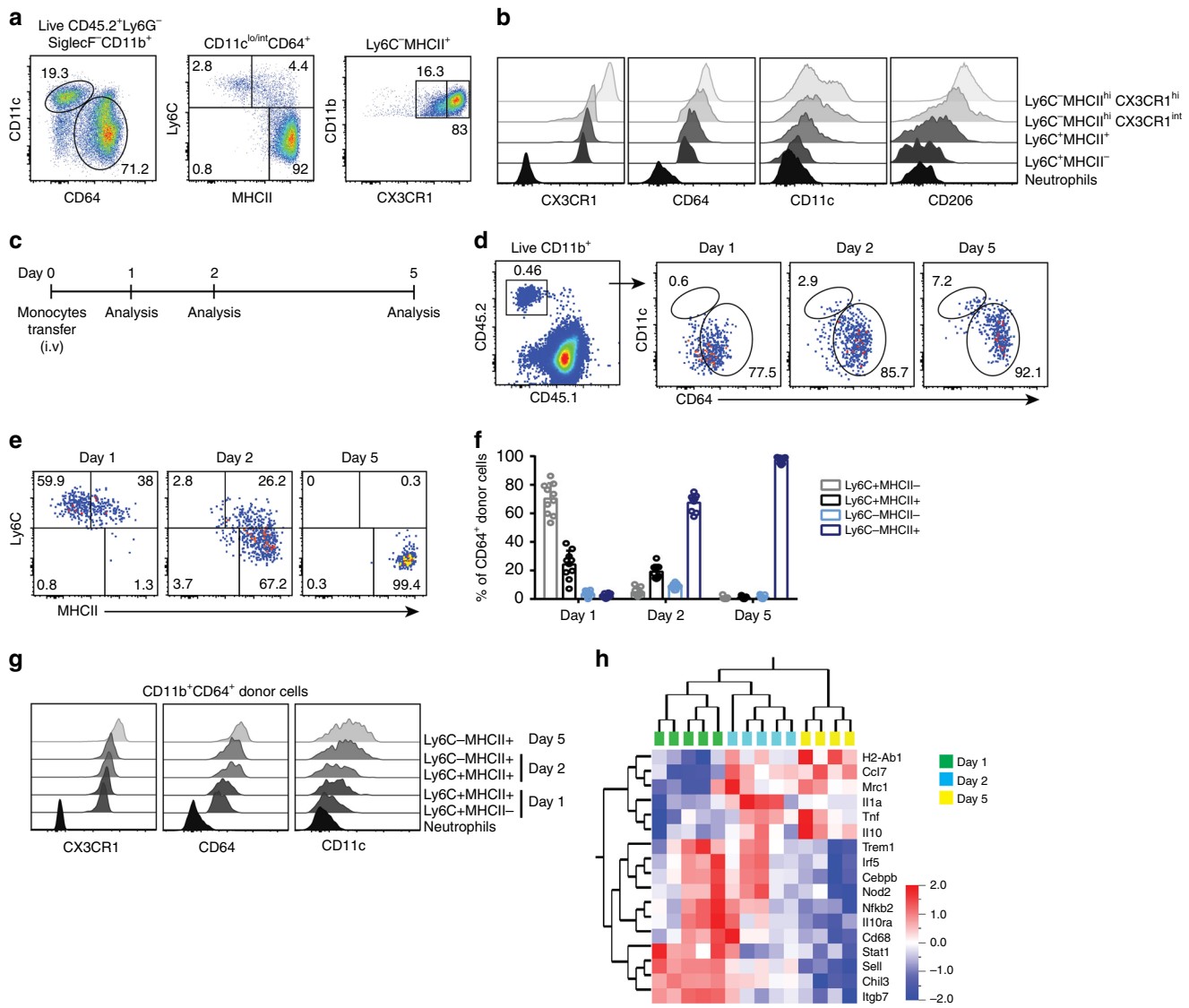

**Fig. 3** Monocytes gradually differentiate into Ly6C⁻MHCIIʰⁱCX3CR1ʰⁱ macrophages in the normal small intestine. **a** Endogenous monocyte and monocyte-derived cells in the small intestinal lamina propria (siLP) were identified as Ly6G⁻SiglecF⁻CD11b⁺CD11cˡᵒ/ⁱⁿᵗCD64⁺ leukocytes (MNPᶜᴰ⁶⁴⁺ cells). MNPᶜᴰ⁶⁴⁺ cells were further divided into four subpopulation based on expression of Ly6C, MHCII, and CX3CR1/GFP; namely Ly6C⁺MHCII⁻, Ly6C⁺MHCII⁺, Ly6C⁻MHCIIʰⁱCX3CR1ⁱⁿᵗ, and Ly6C⁻MHCIIʰⁱCX3CR1ʰⁱ. **b** Representative histograms demonstrating expression of CX3CR1, CD64, CD11c, and CD206 on subpopulations shown in **a** and neutrophils (CD11b⁺Ly6G⁺) for comparison. **c** Schematic illustration of the experimental setup for **d–g**. Purified monocytes were adoptively transferred to recipient mice and donor cells were recovered from siLP at day 1, 2, and 5 after transfer. **d** Donor MNPᶜᴰ⁶⁴⁺ cells recovered from siLP of the recipients were identified by congenic markers followed by gating as described in left panel of **a**. **e** Representative FACS plots demonstrating donor MNPᶜᴰ⁶⁴⁺ cells that gradually lost expression of Ly6C and gained expression of MHCII from day 1 to day 5 after transfer. **f** Frequency of Ly6C- and MHCII-defined subsets among donor MNPᶜᴰ⁶⁴⁺ cells in the intestine at the indicated time points. Data were pooled from at least 4 independent experiments with a total of 9–10 mice each. **g** Expression of CX3CR1/GFP, CD64, and CD11c on indicated subsets of donor MNPᶜᴰ⁶⁴⁺ cells at day 1, 2, and 5 post-transfer compared to endogenous neutrophils. Results in **g** are representative of 4 independent experiments each. **h** Heat map showing hierarchal clustering of gene expression profiles of adoptively transferred MNPᶜᴰ⁶⁴⁺ cells. Donor monocyte-derived cells were recovered from siLP of recipient mice at day 1, 2, or 5 after transfer and transcriptomic analysis was performed by nanostring nCounter analysis. Differentially expressed genes were determined by Qlucore Omics explorer with a statistical significance p-value of 0.05. Data were pooled from 2–3 independent experiments with a total of 4–5 mice for each time point

transferred cells acquired MHCII expression and downregulated Ly6C. This differentiation process was continued by day 5 when all transferred cells consistently showed a high level of MHCII expression and complete loss of Ly6C (Fig. 3e, f). Expression of CX3CR1 gradually increased from day 1 to day 5 along with expression of CD64 and CD11c as the donor cells differentiate from Ly6C⁺MHCII⁻ monocytes to Ly6C⁻MHCʰⁱ macrophages

through an intermediary stage of Ly6C⁺MHCII⁺ (Fig. 3g). These data extend the previous findings in colon[11] and dermis[35], and establish a successive series of developmental stages for monocytes in the non-inflamed small intestine.

Flow cytometry based characterization is limited to a rather restricted number of markers and thus may easily overlook important changes in cell phenotypes. To refine the

characterization of monocyte differentiation, we identified a set of 67 genes (Supplementary Table 1) to reflect different developmental stages and functional differences between tolerogenic and proinflammatory monocytes/macrophages[11,12,34]. Genes in our study were carefully selected to recapitulate main transcriptomic differences of the known stages of monocyte-to-macrophage differentiation in the non-inflamed colon[34]. Analysis of these gene transcripts was performed on one hundred FACS-purified MNP$^{CD64+}$ donor monocyte-derived cells 1, 2, and 5 days after transfer by nanostring nCounter gene expression analysis. Differentially regulated genes were determined by using Qlucore Omics explorer with statistical significance p-value cutoff of 0.05. Ten genes were excluded from the analysis (Supplementary Table 1, 2) due to low detection levels in all samples.

Interestingly, transcriptomic profiles showed clear clustering according to the days after transfer, indicating that the time spent in the tissue had a major effect on a broader gene expression profile of adoptively transferred monocytes (Fig. 3h). Consistent with our flow cytometric analysis, H2-Ab1 encoding MHCII was upregulated in monocyte-derived cells from day 1 to day 5 post-transfer. In addition, Il10 and Mrc1, genes encoding IL-10 and CD206, respectively, were upregulated (Fig. 3h). Both IL-10 and CD206 are considered to have anti-inflammatory properties in colonic macrophages[10,36]. Similarly, and consistent with other studies conducted in colon[11,12], Tnf and Ccl7 increased in expression from day 1 to day 5. In contrast, several genes and transcription factors associated with immune recognition and activation (Nod2, Trem1, CD44, CD68, Stat1, Nfkb2, and Irf5) were downregulated upon maturation from day 1 to day 5 (Fig. 3h; Supplementary Fig. 5c, 6a). In addition, genes encoding adhesion and homing molecules (Sell and Itgb7; encoding CD62L and β7-integrin, respectively) were highest in donor monocytes at day 1 and their expression diminished upon maturation to macrophages. Taken together, this gene expression analysis confirms the currently held view that during homeostasis in the gut monocytes gradually differentiate into macrophages that exhibit anti-inflammatory properties.

**Inflammation immediately redirects monocyte differentiation.** We next extended our studies to examine monocyte differentiation in the context of intestinal inflammation. One day after surgical gut manipulation, the endogenous MNP$^{CD64+}$ population in the inflamed small intestine can be divided into three subsets according to Ly6C and MHCII expression, which is comparable to homeostatic conditions, with the exception of massive accumulation of Ly6C expressing monocytes (Supplementary Fig. 2d, 3d). However, subtle phenotypic differences were apparent when expression of CX3CR1 and CD64 on Ly6C$^+$ monocytes was considered. In manipulated recipients, CX3CR1 expression on small intestinal Ly6C$^+$ monocytes was consistently lower whereas CD64 expression was higher as compared to non-manipulated mice (Supplementary Fig. 3b, c). This observation is consistent with previous studies reporting accumulation of Ly6C$^+$CX3CR1$^{int}$ cells in the inflamed colon[11,12]. However, at day 3 post-manipulation, in contrast to the MNPs under homeostatic conditions, we noted two subsets of Ly6C$^-$CX3CR1$^{int}$ MNPs$^{CD64+}$ (Fig. 4a, b): a population of MHCII$^{hi}$CX3CR1$^{int}$ cells, also described in inflamed colon[11,12] as well as a population characterized by intermediate expression of MHCII not previously reported in other models[11,12,22]. Expression levels of CX3CR1 on this Ly6C$^-$MHCII$^{int}$ population were slightly higher than on the Ly6C$^+$MHCII$^-$ and Ly6C$^+$ MHCII$^{int}$ populations, but lower as compared to the Ly6C$^-$ MHCII$^{hi}$ macrophages, suggesting that these cells may be an

intermediate differentiation stage in the inflamed small intestine (Fig. 4b).

To precisely determine the interrelation of intestinal monocyte-derived cells, we used the monocyte transfer system in recipients with established gut inflammation. Recipients underwent small intestinal manipulation one day before monocyte transfer. The phenotype of adoptively transferred monocytes was characterized 1, 2, or 5 days post-transfer (Fig. 4c) by multiparameter flow cytometry and nanostring gene expression analysis. Virtually all donor CD11b$^+$ cells expressed high levels of CD64 at all time points (Fig. 4d). However, our phenotypic analysis revealed differences in monocyte differentiation compared to homeostasis. At day 1, transferred monocytes showed no apparent differences in surface marker expression compared to homeostasis, except for reduced expression of CX3CR1 and slightly higher expression of CD64. However, more notable phenotypic differences became apparent by day 2 after transfer. At day 2, transferred cells progressed from a Ly6C$^+$MHCII$^-$ to a Ly6C$^-$MHCII$^{int}$ phenotype expressing intermediate levels of CX3CR1 (Fig. 4e–g), a population reminiscent of the Ly6C$^-$MHCII$^{int}$ endogenous cells we observed in inflamed small intestine. By day 5, all transferred cells downregulated Ly6C and the majority increased expression of MHCII (Fig. 4e), although levels were still lower as compared to non-manipulated counterparts (Fig. 3e). In addition, the majority of the donor monocyte-derived cells at day 5 expressed intermediate level of CX3CR1 (Fig. 4g; Supplementary Fig. 5a). Notably, the phenotype of adoptively transferred monocytes recovered 1, 2, or 5 days post-transfer into manipulated WT recipients recapitulated the phenotypes observed in CCR2-deficieint recipients (Supplementary Fig. 5b). This shows that monocytes still undergo gradual phenotypic changes upon migration into the inflamed gut. However, the differences in surface marker expression as compared to homeostatic condition suggest that they follow a different developmental continuum in the inflamed small intestine.

To further characterize monocyte development in inflamed small intestine, we performed transcriptomic analysis. In inflamed small intestine, similarly to the situation during homeostasis, monocytes gradually changed their transcriptional profile from day 1 to day 5 (Fig. 4h). Siglec-1 (CD169) was the only gene tested that was significantly upregulated from day 1 to day 5 in manipulated small intestine. CD169 expressing macrophages have been suggested to enhance inflammation by producing CCL8, a chemokine required for monocyte influx[37]. In contrast, several genes associated with inflammatory processes gradually declined from day 1 to day 5, and some genes that increased expression under homeostatic condition were not differentially expressed in inflamed small intestine (Fig. 4h; Supplementary Fig. 5c). This shows that monocytes in the inflamed small intestine undergo a different differentiation program as compared to homeostasis.

To precisely characterize the differences in monocyte differentiation between non-inflamed and inflamed small intestine, we directly compared the transcriptomes of donor monocyte-derived cells in both conditions. Donor monocyte-derived cells showed distinct expression profiles according to the state of the recipients at all time points (Fig. 5a). In addition, principal component analysis (PCA) revealed that the greatest differences between adoptively transferred monocytes were detected as early as one day after transfer (Fig. 5b). These differences reduced over time, and by day 5 after transfer, the expression profile of monocytes in inflamed and non-inflamed small intestine converged (Fig. 5b). This observation was also supported by a pairwise clustering of monocyte expression

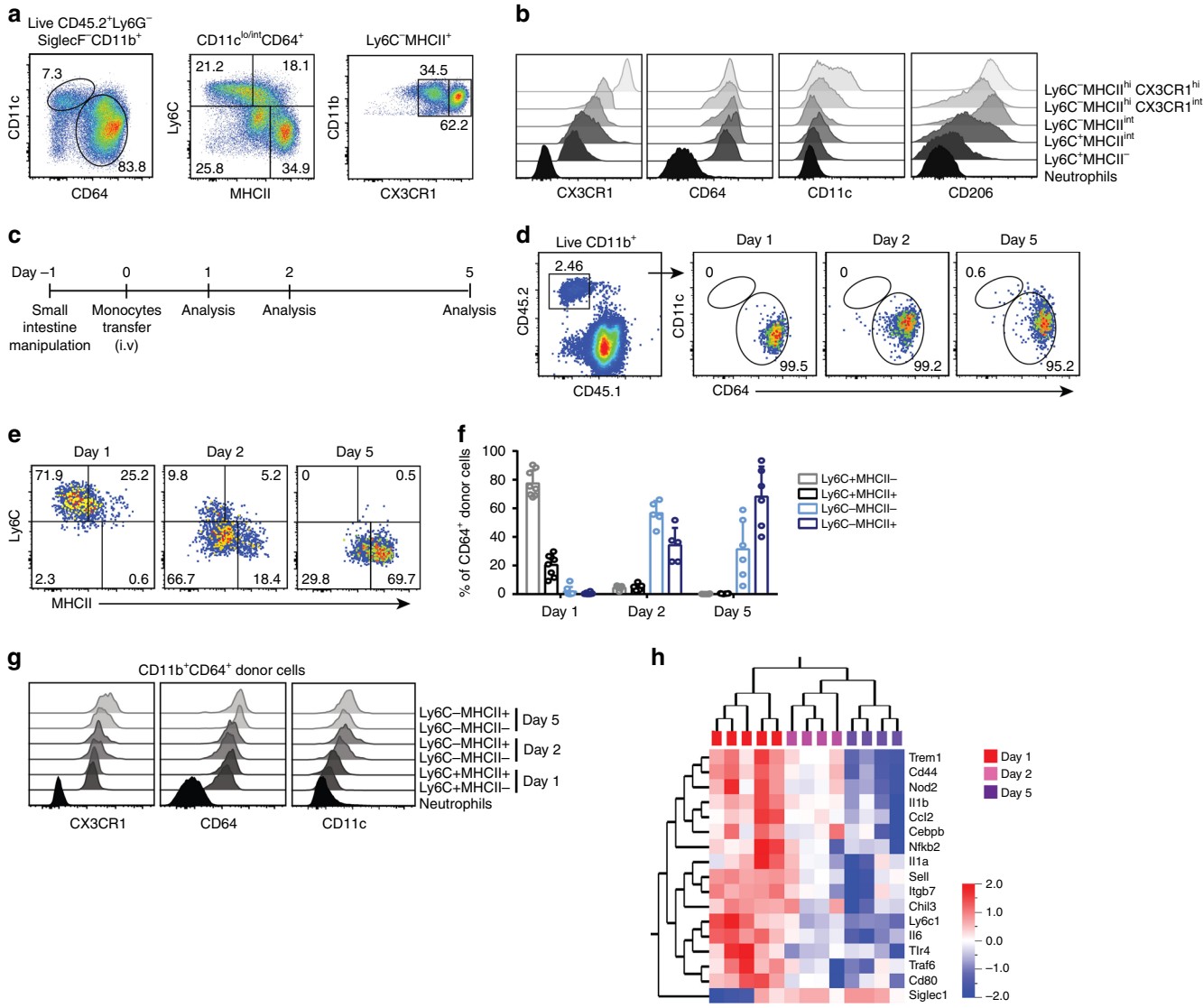

**Fig. 4** Inflammation alters monocyte differentiation program in the small intestine. **a** Small intestine were surgically manipulated to induce inflammation in the small intestine and 3 days after manipulation, intestinal Ly6G$^-$SiglecF$^-$CD11b$^+$CD11c$^{lo/int}$CD64$^+$ leukocytes (MNP$^{CD64+}$ cells) were divided into five subpopulations based on expression of Ly6C, MHCII, and CX3CR1/GFP; namely Ly6C$^+$MHCII$^-$, Ly6C$^+$MHCII$^{int}$, Ly6C$^-$MHCII$^{int}$, Ly6C$^-$MHCII$^{hi}$CX3CR1$^{int}$ and Ly6C$^-$MHCII$^{hi}$CX3CR1$^{hi}$. **b** Representative histograms demonstrating expression of CX3CR1, CD64, CD11c, and CD206 on subpopulations shown in **a** and neutrophils (CD11b$^+$Ly6G$^+$) for comparison. **c** Schematic illustration of the experimental setup for **d**–**g**. Purified monocytes were adoptively transferred to recipient mice that underwent surgical manipulation of the small intestine one day before the transfer. Donor cells were recovered from small intestinal lamina propria (siLP) 1, 2 and 5 days after transfer. **d** Transferred MNP$^{CD64+}$ cells recovered from siLP of the manipulated recipients were identified by congenic markers followed by gating as described in left panel of **a**. **e** Representative FACS plots demonstrating gradual phenotypic changes of donor MNP$^{CD64+}$ cells over time, with respect to their expression of Ly6C and MHCII. **f** Frequency of Ly6C- and MHCII-defined subsets among donor MNP$^{CD64+}$ cells in the intestine at the indicated time points. Data were pooled from at least three independent experiments with a total of 5–8 mice. **g** Expression of CX3CR1/GFP, CD64, and CD11c on donor MNP$^{CD64+}$ cells at day 1, 2, and 5 post-transfer compared to endogenous neutrophils. Results in **g** are representative of at least three independent experiments each. **h** Heat map showing hierarchal clustering of gene expression profiles of adoptively transferred MNP$^{CD64+}$ cells. Donor monocyte-derived cells were recovered from siLP of manipulated recipients at day 1, 2, or 5 after transfer and transcriptomic analysis was performed by nanostring nCounter analysis. Differentially expressed genes were determined by Qlucore Omics explorer with a statistical significance $p$-value of 0.05. Data were pooled from 2–3 independent experiments with a total of 4–5 mice for each time point

profiles at day 1, 2, and 5: whereas one day after transfer 21 genes were differentially expressed between manipulated and non-manipulated recipients, by day 2 this number was reduced to 19 and by day 5 only 10 genes were differentially expressed (Fig. 5c–e). Notably, this comparison revealed several genes that were consistently higher at all three time points in the inflamed intestine, such as *Il1a, Il23a, Ccl2, Cxcl1, Cd14,* and *Arg1,* whereas other genes showed time point restricted

differences in expression. Interestingly, the expression profile of purified BM monocytes, i.e. the input population for the transfer, clustered closely with donor monocytes recovered one day after transfer from healthy small intestine but was markedly different to those isolated from inflamed intestine (Fig. 5b). These data point to an immediate and profound transcriptional rewiring of monocytes as soon as they enter the inflamed intestine which was not revealed by flow cytometric analysis.

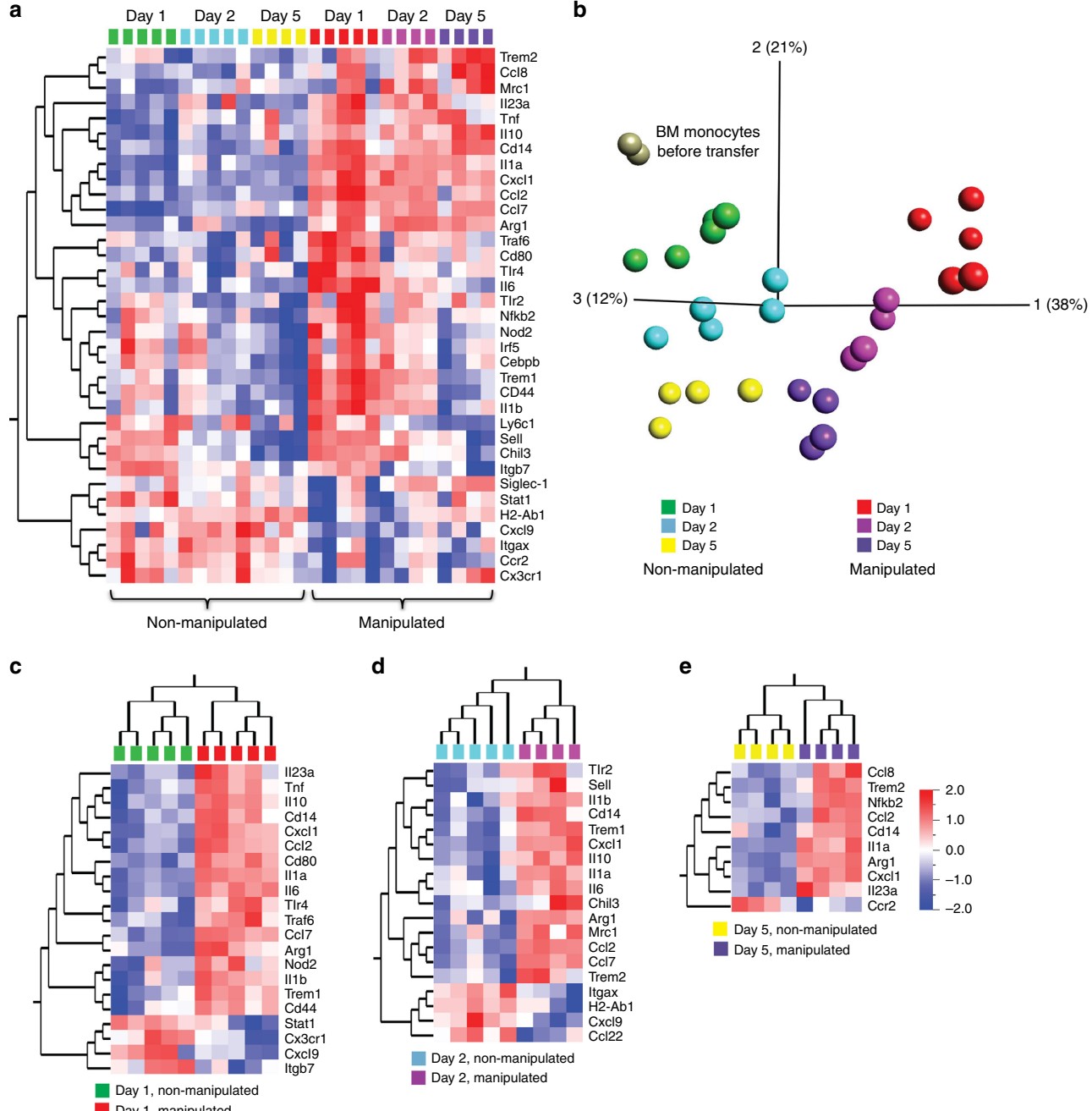

**Fig. 5** Inflammation triggers transcriptomic changes in monocyte immediately upon entry into the small intestine. **a** Heat map demonstrating gene expression profiles of adoptively transferred CD11b+CD11c$^{lo/int}$CD64+ (MNP$^{CD64+}$) cells recovered small intestinal lamina propria (siLP) of non-manipulated and manipulated recipients at day 1, 2, and 5 after transfer. **b** Principal component analysis (PCA) representing transcriptomic profiles of donor BM monocytes before transfer and adoptively transferred MNP$^{CD64+}$ cells recovered from siLP of non-manipulated and manipulated recipients at day 1, 2, and 5 post-transfer. **c–e** Heat map showing hierarchal clustering of gene expression profiles of donor MNP$^{CD64+}$ cells recovered from the siLP at day 1 (**c**), day 2 (**d**), and day 5 (**e**) after transfer comparing non-manipulated and manipulated recipients. Transcriptomic analysis was performed by nanostring nCounter analysis and differentially expressed genes were determined by Qlucore Omics explorer with a statistical significance p-value of 0.05. Data were pooled from 2–3 independent experiments with a total of 2–5 mice as indicated in each section of the figure

The factors the shape intestinal macrophage differentiation and adaptation to intestinal inflammation are largely unknown. However, interesting candidates might be found among the genes that show differential regulation between inflamed and non-inflamed conditions. As an example of such a factor, we selected Trem1 for further functional studies. Trem1 shows gradual downregulation between day 1 and day 5 after transfer and

additionally exhibits a differential expression between inflamed and non-inflamed intestine (Supplementary Fig. 5c, Supplementary Fig. 6a). To explore the functional relevance of Trem1 in intestinal macrophage differentiation, we transferred Trem1-deficient monocytes into manipulated and non-manipulated recipients and compared their transcriptomic signature to WT monocytes recovered from manipulated and non-manipulated

recipients. The nanostring analysis revealed no major differences between Trem1-deficient and WT bone marrow monocytes (out of the 67 genes analyzed only two genes, IL23a and CXCL9, showed significant differences in their expression). This suggests that Trem1 did not have a major effect on the phenotype of bone marrow monocytes with respect to these genes (but does not exclude potential effects of Trem1 on the overall monocyte transcriptome). In contrast adoptively transferred Trem1-deficient monocytes were substantially different from their WT counterparts (Supplementary Fig. 6b). Indeed, Trem1-deficient monocytes showed a 'split signature' faithfully recapitulating the upregulation of some but not all genes upregulated in WT monocytes entering the small intestinal environment, indicating that Trem1 indeed plays a role in directing the differentiation of monocytes to macrophages in the gut.

**Tissue-specific signals instruct local monocyte adaptation.** Intestinal inflammation increases monocyte output from the BM[38] and promotes their massive influx into the intestinal tissue (Supplementary Fig. 2d). Our data show that recruitment rate of monocytes to non-inflamed tissues such as liver and spleen is not affected by intestinal inflammation. However, it is less clear whether the monocytes which are still recruited to these tissues are affected transcriptionally by the intestinal inflammation. Analysis of adoptively transferred monocytes re-isolated at day 1 from liver, spleen and BM revealed few differentially regulated genes comparing manipulated and non-manipulated recipients (Fig. 6a). Notably, the differentially expressed proinflammatory cytokines and chemokines in inflamed small intestine at day 1 appeared unaffected in liver, spleen and BM of manipulated recipients (Fig. 6a). This shows that gut manipulation-induced inflammation exerts a largely intestine-specific impact on monocyte transcription programing.

Consistently, we found that small intestine, liver, spleen, and BM differentially instructed the gene expression profile of incoming monocytes. Similar to the situation in the small intestine, donor monocytes adapted their phenotype in a highly tissue-specific manner in the liver, spleen, and BM of non-manipulated recipient mice one day after transfer (Fig. 6b). Whereas some genes, such as H2-Ab1 and Itgax (encoding MHCII and CD11c, respectively) were consistently upregulated upon recruitment into all four tissues, other genes showed a tissue-specific pattern of regulation (Fig. 6b). In particular, appearance of transferred monocytes in small intestine and liver was accompanied by upregulation of genes associated with activation and proinflammatory features but not in the spleen and BM (Fig. 6b). Indeed, comparison of expression profiles between all four tissues analyzed, showed differential gene regulation patterns. Since our expression analysis is focused on candidate genes, our analysis might miss interesting differences in the expression profile of monocytes recruited into different organs. However, for the list of candidate genes analyzed here, the overall pattern of regulation was more similar for monocytes in small intestine and liver as compared to spleen and BM (Fig. 6c). Interestingly, small intestine and liver still clustered together in recipients that underwent gut manipulation before adoptive transfer (Fig. 6d). These findings indicate that tissue-specific signals play an important role in shaping monocytes as they enter a particular tissue and that such instructive signals might be more similar in small intestine and liver as compared to spleen and bone marrow.

## Discussion
Monocyte-derived macrophages in the intestine have different phenotypes and functions during homeostasis and inflammation.

How this phenotypic and functional heterogeneity is established is not fully understood. Monocyte-to-macrophage differentiation in normal colon and dermis is considered a progressive process with differentiating monocytes passing through distinct phenotypic stages to become mature intestinal macrophages[11,34,35]. Here we performed adoptive monocyte transfers to dissect the stages and kinetics of monocyte-to-macrophage differentiation in the small intestine under homeostatic and inflammatory conditions. As a particular strength of this approach, the monocyte transfer system allows to resolve the kinetics of the differentiation process and defines the precise origin of the analyzed cell types. To the best of our knowledge, this is the first study to report transcriptional profiling of adoptively transferred monocytes in the small intestine.

We show that in normal small intestine, like in colon and dermis, monocytes undergo gradual phenotypic and transcriptional changes. Already as early as 1 day after transfer, donor monocytes in small intestine showed an expression profile distinct from monocytes in spleen and BM. This finding confirms previous suggestions of tissue-specific signals shaping the differentiation of incoming monocytes to adopt tissue-specific phenotypes and functions[11,34,35]. The precise signals that mediate such tissue-specific differentiation remain to be elucidated. At later time points, as donor monocytes differentiated further in the small intestine, they displayed stepwise transition to mature macrophages, similarly to that observed in the colon and dermis[11,34,35]. Comparative transcriptional profiling of endogenous monocytes/macrophages between colon and dermis showed that changes accompanying monocyte maturation are highly tissue-specific[34]. In our study, expression of Tnf and Ccl7 increased as monocytes differentiated in the small intestine, similarly to macrophages in normal colon[10–12,34]. TNF has been reported to have dual role of pro-and anti-inflammatory activity[39,40]. Also similarly to the situation observed in normal colon[11,34], H2-Ab1, Mrc1, and Il10 (encoding MHCII, CD206 and IL-10) were upregulated as monocyte differentiated to macrophages over time in the small intestine. Intestinal macrophages constitutively produce high level of IL-10 [10,11,41], which is crucial for the expansion of Treg cells and induction of oral tolerance in the intestine[4]. Thus, our data extend the concept of progressive tissue-specific monocyte differentiation to the normal small intestine.

Preceding local monocyte-to-macrophage differentiation, monocytes need to home to the small intestine and indeed modulation of monocyte homing has therapeutic potential. Here we used a competitive adoptive transfer approach to show that CCR2 and β7-integrin, but not CCR9, are required for monocyte migration from the blood into the small intestine and colon under homeostatic conditions. Monocytes egress from BM in a CCR2-dependent manner, and CCR2 deficiency markedly reduced their abundance in blood and peripheral tissues (reviewed in[42]). In addition, previous observations in BM chimeras reconstituted with mixed WT:CCR2[-/-] BM indicated that CCR2 also mediates homing of circulating monocytes into both small intestine and colon[20,22]. In line with this, our data demonstrate that CCR2 is also essential for the recruitment of circulating monocytes to both the small intestine and colon. Interestingly, recruitment of circulating monocytes to other tissues such as liver and spleen was not affected by CCR2 deficiency. In addition, we found that trafficking of CCR2[−/−] monocytes to the small intestine was also diminished during inflammation. This finding suggests that targeting CCR2 on circulating monocytes may selectively reduce their trafficking to the inflamed intestine and may be a promising approach for treatment of inflammatory diseases, such as IBD, without affecting monocyte trafficking to other tissues. Besides the tissue-specific role of CCR2, our data also demonstrated a

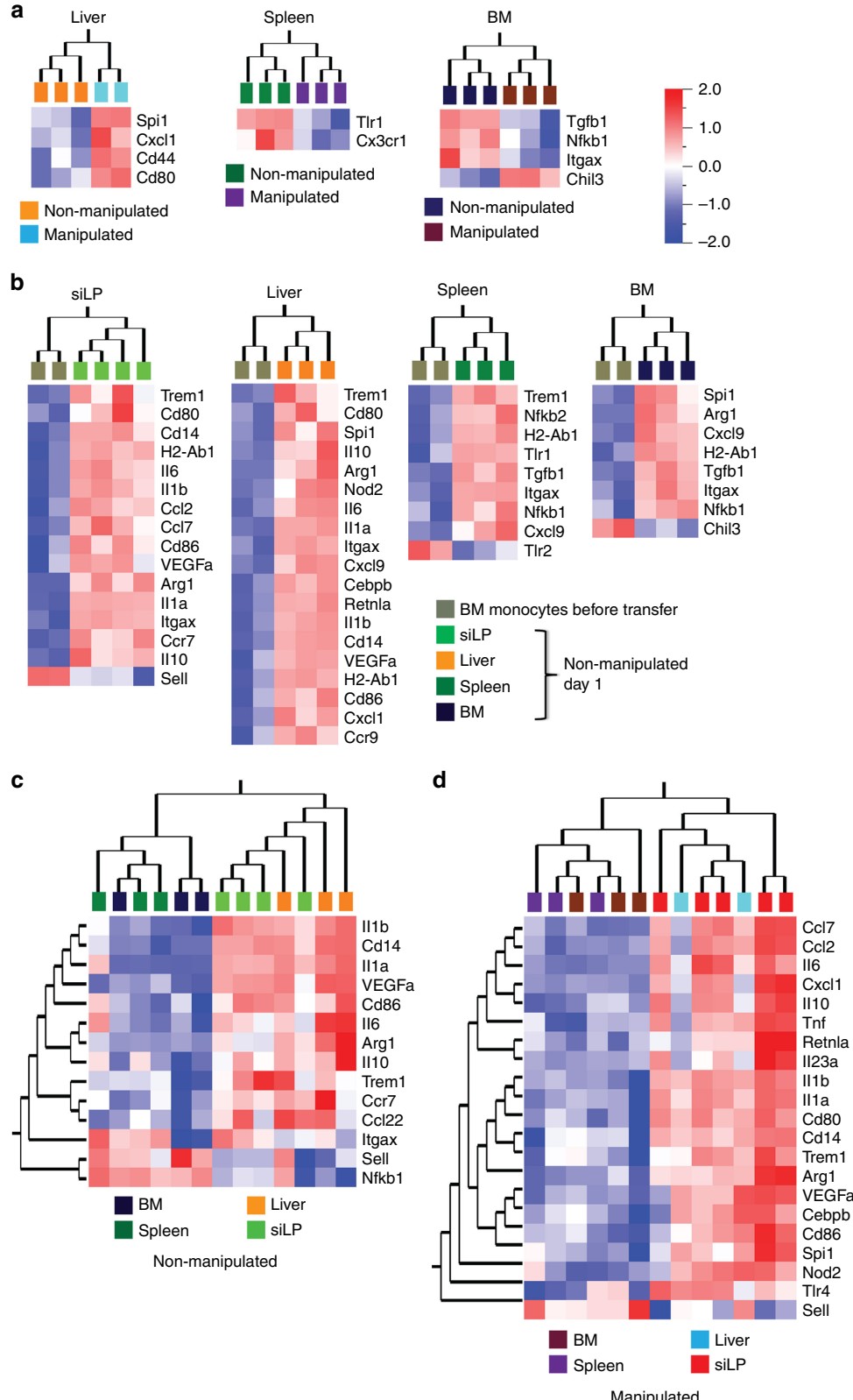

**Fig. 6** Tissue-specific signals instruct local monocyte adaptation. **a** Heat map showing gene expression profiles of donor monocytes recovered one day after transfer from liver, spleen, and BM comparing non-manipulated and manipulated recipients. **b** Gene expression profiles comparing donor BM monocytes before transfer with donor monocytes recovered from small intestine, liver, spleen, or BM of non-manipulated recipient at day 1 after transfer. **c**, **d** Comparison of gene expression profile of donor monocytes recovered at day 1 after transfer between small intestine, liver, spleen and BM of non-manipulated (**c**) and manipulated (**d**) recipients. Transcriptomic analysis was performed using nanostring nCounter analysis and differentially expressed genes were determined by Qlucore Omics explorer with a statistical significance *p*-value of 0.05. Data were pooled from 2–3 independent experiments with a total of 2–4 mice as indicated in each section of the figure

previously unreported role of β7-integrin in homing of circulating monocytes to the small intestine and colon under homeostatic conditions. In contrast, we found that β7-integrin had no detectable role in monocyte homing to the small intestine during inflammation.

The differential role of β7-integrin in homing of circulating monocytes to the small intestine in a context-dependent manner is not clear. One possible explanation could be that different subsets of monocytes are recruited into the inflamed intestine. However, flow cytometric analysis of several surface markers indicated homogeneity of the BM and blood monocyte populations (Supplementary Fig. 4). Therefore, unperturbed recruitment of integrin β7-deficient monocytes into the inflamed intestine hints at differential mechanisms mediating monocyte recruitment during homeostsis and inflammation.

Indeed, we found that *Itgb7* is downregulated in monocytes entering small intestine under inflammatory compared to non-inflamed conditions, which could explain why β7-deficient monocytes had no disadvantage in migrating during inflammation. Our observations contrast with published observations that β7-deficient mice have reduced accumulation of monocytes in the colon in an experimental colitis model[25]. The use of different models of inflammation and putative differences between small and large intestine might explain these contrasting observations.

Crucially, differences between homeostasis and inflammation were not limited to monocyte homing. Inflammation also fundamentally changed the kinetics and modalities of local monocyte differentiation. While initially, monocytes in inflamed and non-inflamed small intestine showed no substantial phenotypic differences, the appearance of a population of Ly6C−MHCII^{int} cells in the inflamed gut at day 2 after transfer suggests that monocytes entering the small intestine during inflammation adopt a distinct differentiation pathway. In previously published work, Ly6C−MHCII^{int} cells have been reported in the muscularis externa of the small intestine after gut manipulation[29] but have not been detected in the lamina propria of colitic mice[11,22]. Thus, at present it is unclear whether presence/absence of the Ly6C−MHCII^{int} cells corresponds to unique tissue localization (small versus large intestine, lamina propria versus muscularis) or differences in inflammation models (small intestinal surgical manipulation or experimental colitis). Differences detectable by flow cytometry remained at day 5 after adoptive transfer. Notably, in inflamed small intestine we observed an abundant population of CX3CR1^{int} macrophages outnumbering the CX3CR1^{hi} macrophages characteristic of the healthy gut. CX3CR1^{int} macrophages were described in the inflamed colon, and it was suggested that they might represent an intermediate stage of monocyte differentiation which is blocked from progressing to the CX3CR1^{hi} macrophages by inflammatory signals[11]. However, transcriptomic analysis suggests an entirely different scenario: that monocytes infiltrating the inflamed small intestine adopt an alternative differentiation program from the earliest time points. Transcriptional profile of monocytes in inflamed recipients diverged as early as one day after transfer and was characterised already at this time point by higher expression of proinflammatory mediators as compared to their counterparts in normal gut. This suggests that, while monocytes entering the non-inflamed and inflamed small intestinal have a superficially similar Ly6C+MHCII− surface phenotype, they already exhibit profound differences in their developmental programing. Consequently, inflammation does not seem to block monocyte differentiation, but rather instructs monocytes to adopt an inflammation-specific expression profile immediately upon entry into tissue. Similarly, transcriptomic profile

observed at later stages after transfer revealed dynamic changes in the cells that were not reflected by major changes in surface phenotype. Interestingly, transcription signatures of donor monocyte-derived cells in inflamed small intestine became more similar to the normal situation at later time points. We speculate that this may reflect resolution of inflammation in our model. If this was the case, our results would indicate that monocytes that had adapted their developmental program to an inflammatory environment still retain functional plasticity and can change to a less inflammatory phenotype.

It has been suggested that intestinal infection may alter the functional differentiation of monocytes prior to egress from the BM[43]. In our system, transferred monocytes derive from non-manipulated donors, further emphasizing that it is the local environmental cues that control monocyte differentiation in the inflamed intestine. In addition, BM monocytes from donors with established small intestinal inflammation developed into a phenotype characteristic of the homeostatic condition when transferred into healthy hosts (Supplementary Fig. 7). The precise tissue-and context-specific factors by which monocytes are instructed to adapt to a particular local microenvironment remain to be investigated. In previous work TGF-βR signaling has been described to contribute to intestinal macrophage differentiation including regulation of CX3CR1 expression[34]. Here, we identified CX3CR1, IL-10R, and Trem1 to be differentially regulated between normal and inflamed small intestine. Consistently, we and others have demonstrated that IL-10R and CX3CR1 deficiency in intestinal macrophages can cause intestinal inflammation and impaired oral tolerance, respectively[4,44], and Trem1 upregulation in intestinal monocytes/macrophages during inflammation has been reported to promote production of proinflammatory mediators that amplify the inflammatory response in both, DSS and T-cell mediated colitis models, as well as in human IBD patients[13,45,46]. Interestingly, we found that Trem1-deficiency affected the expression profile of transferred monocytes in manipulated and non-manipulated recipients. In synergy with pattern recognition receptor signalling, Trem1 can amplify inflammatory pathways[47], indicating that Trem1 regulation contributes to shaping the intestinal macrophage phenotype in the small intestine. Curiously, Trem1-deficient monocytes transferred into manipulated recipients failed to upregulate several genes upregulated in WT monocyte and further studies analyzing a potential function of Trem1 in intestinal monocyte-to-macrophage differentiation might reveal interesting regulatory mechanisms in this pathway.

Our characterization of monocyte-to-macrophage differentiation in the small intestine has unexpectedly revealed that inflammation profoundly modifies monocyte properties and their developmental pathway immediately upon entry into gut. Since intestinal inflammation also elicits a strong, rapid CCR2-mediated increase in monocyte recruitment, newly migrating monocytes might act as a potential key proinflammatory cell type and not only as precursors of inflammatory macrophages. This potential role and the interplay of recently recruited monocytes with intestine-specific factors such as intestinal microbiota, dietary proteins and metabolites[48–50], need to be considered by future studies.

## Methods

**Mice**. C57BL/6, CX3CR1^{gfp/+[21]}, CCR2^{−/−[51]}, Itgb7^{−/−[52]} and CCR9^{−/−[53]}, Trem1^{−/− [54]} mice were bred and maintained on C57BL/6 background under specific-pathogen free conditions at the central animal facility of RWTH Aachen University. Experiments were approved by North Rhein-Westphalia State Agency for Nature, Environment and Consumer Protection (Landesamt fur Natur, Umwelt und Verbraucherschutz Nordrhein-Westfalen, LANUV) or the Niedersächsisches Landesamt für Verbraucherschutz und Lebensmittelsicherheit (LAVES). All

experiments were performed in accordance with relevant local guidelines and regulations.

**Small intestinal manipulation**. To induce inflammation in the small intestine, a model of surgical gut manipulation was performed as previously described[27–29]. Mice were anesthetized using a mixture of ketamine and xylazine. A midline abdominal incision was made and the small intestine was exteriorized and spread out onto gauze tissue pad moistened with phosphate buffered saline (PBS) (Gibco). Both sides of the intestine were then gently flattened by the smooth side of dressing blunt forceps and put back to the peritoneal cavity. The peritoneum was closed with simple interrupted suture (MARLIN Violett) followed by skin closure with surgical clips (Cellpoint scientific). Manipulated mice, once recovered from anesthesia, were on drinking water containing Novalgin for the first three days post-surgery. Mice were then sacrificed 2, 3, or 6 days after manipulation. Non-manipulated mice did not undergo any surgical procedure.

**Cell isolation**. Cells from the intestinal lamina propria were isolated as described previously[34]. Briefly, after the fat, mesentery and Peyer's patches were removed, the intestine was opened longitudinally, the luminal content flushed with PBS/3% FBS and the intestine cut into approximately 5 cm long pieces. The tissue was then incubated $3 \times 15$ min in pre-warmed HBSS (Gibco) containing 10%FBS and 2 mM EDTA (Carl-Roth) in a 37 °C shaking incubator, then incubated in pre-warmed RPMI 1640 (Gibco) with 10%FBS, 1 mg/ml Collagenase VIII (Sigma–Aldrich) and 30 μg/ml DNase I (Roche) for 20–30 min in a 37 °C shaking incubator. Cells from colonic lamina propria were also isolated in a similar procedure except that the tissue was digested in RPMI/10%FBS supplemented with 0.85 mg/ml Collagenase V (Sigma–Aldrich), 1.25 mg/ml Collagenase D (Roche), 1 mg/ml Dispase (Gibco), and 30 μg/ml DNase I (Roche). To isolate spleen cells, spleen was smashed with a syringe plunger and erythrocytes were lysed. For BM cell isolation, femurs and tibias were flushed and erythrocytes were lysed. For liver cell isolation, liver was chopped finely into small pieces and then digested by incubating in RPMI/10%FBS with 0.5 mg/ml Collagenase VIII and 30 μg/ml DNase I in a 37 °C shaking incubator for 40–50 min before erythrocyte lysis.

**Histology**. Sections excised from the jejunum were fixed in 4% paraformaldehyde (Carl-Roth), embedded in paraffin (Carl-Roth), and stained with hematoxylin and eosin (Sigma–Aldrich) for microscopic assessment of intestinal architecture.

**Monocyte enrichment**. BM CD11b+Ly6C+ monocyte enrichment was performed using the monocyte isolation kit (Miltenyi Biotec) according to manufacturer's instructions. Briefly, BM cells were suspended in MACS buffer and Fc receptor (FcR) blocking solution was added and incubated with a cocktail of biotinylated primary monoclonal antibodies at 2–8 °C for 5 min to label non-targets cells. Cells were washed once and secondary anti-biotin monoclonal antibodies conjugated to Microbeads were then added to cell suspension. After 10 min of incubation at 2–8 °C, cell suspensions were run in autoMACS ProSeparator (Miltenyi Biotec) and depletion program ''Depletes'' was used throughout all experiments to deplete non-target cells.

**Adoptive monocyte transfer**. Purified BM CD11b+Ly6C+ monocytes from donor mice were intravenously transferred into recipient mice (1.5–2.5 × 10^6 cells/mouse). For competitive adoptive monocyte transfer, purified BM monocytes from CD45.2+ WT and either CCR2, β7-integrin or CCR9 knockout mice were labeled with eFlour 450 and eFlour 670 (eBioscience) and mixed at a 1:1 ratio. Labeled cells were intravenously transferred into CD45.1+CCR2−/− non-manipulated or manipulated recipients. Along with these experiments, monocytes from WT donors were labeled with either eFlour 450 or eFlour 670, mixed at a 1:1 ratio and competitively transferred to recipient mice as control. One day after transfer, donor monocytes were recovered from small intestine, colon, liver, spleen and BM, and the ratio of gene-deficient to WT donor monocytes in respective tissues was determined by flow cytometry. For monocyte differentiation experiments, purified BM monocytes from CD45.2+ WT or CX3CR1gfp/+ donors were intravenously injected into non-manipulated or manipulated CD45.1+ CCR2-/- recipients. Donor monocyte-derived cells were then re-isolated from small intestines of recipient mice at 1, 2 or 5 days after transfer for flow cytometric and transcriptomic analysis.

**Flow cytometry and cell sorting**. Flow cytometric studies were performed using LSRFortessa (BD) and analyzed by FlowJo VX software (Treestar). FACSAria II (BD) was used to sort cells. For staining, single-cell suspensions were resuspended in PBS/3%FBS and incubated in 5% rat serum to block non-specific binding. Surface marker staining was performed using fluorochrome-conjugated mono-clonal antibodies listed in Supplementary Table 3. DAPI or 7AAD (Biolegend) were used to stain dead cells depending on the experiment. The FACS gating strategies used to define cell populations are shown in Supplementary Figure 8.

**Gene expression analysis by nanostring nCounter technology**. nCounter single-cell gene expression analysis protocol (nanostring technologies inc.) was used to determine of transcriptional profile of monocytes and adoptively transferred

monocyte-derived cells. 67 genes of interest and five house-keeping genes selected to be included in this analysis (supplementary tables 1 and 2). One hundred transferred cells were sorted into a PCR tube (VWR) containing lysis buffer. RNA was reverse-transcribed to cDNA immediately after sorting using the SuperScript® VILO™ Master Mix (Invitrogen). Multiplexed target enrichment (MTE) was performed by adding a pool of gene-specific MTE primers (Integrated DNA Technologies (IDT) and TaqMan® PreAmp MasterMix (Applied Biosystems) to cDNA samples and run for 16-cycles of amplification on a peqSTAR thermal cycler (peqlab). This allows pre-amplification of target nucleic acid sequences present in the very low amount of cDNA. Forward and reverse primers used in this assay are listed in the supplementary table 1. To evaluate whether each pre-amplified DNA sample was suitable for hybridization, β-actin gene expression level was measured by qPCR as described below. Pre-amplified DNA was then hybridized with target specific oligonucleotide probe pairs that bind to barcoded tags (TagSet, ELE-P1TS-072, nanostring technologies inc, supplementary table 2). Hybridization was performed according to manufacturer's instructions by incubating the reaction tube containing samples, probes and Tagset at 67 °C for 19 h. After 19 h of incubation, samples were run in nCounter Prep-station (nanostring technologies inc.) to remove unbound probes and tags, and immobilize hybridized sequences on sample cartridge surface. Sample cartridges were analyzed by nCounter digital analyzer (nanostring technologies inc.). Relative expression level of genes across samples was analyzed after normalizing to the house-keeping gene expression and presented as a heat map and three-dimensional PCA.

**β-actin expression by real-time quantitative PCR**. DNA from MTE pre-amplified samples was taken and gene expression of β-actin was determined by CFX96 RT-PCR using the SYBR Green mix kit (Bio-Rad laboratories). Both forward (GATGCCCTGAGGCTCTTTTCC) and reverse (TGGCATA-GAGGTCTTTACGGATGT) primers (Sigma–Aldrich) that bind to β-actin sequences within the MTE pre-amplified region were used at final concentration of 500 nM for each primer. The quantification cycle (Cq) of β-actin was determined from each sample, in duplicate, after 30 cycles of amplification by Bio-rad CXF manager software (Bio-Rad laboratories). Melting curve was generated in all PCRs to check for presence of non-specific products. Samples with a Cq of more than 14 were exempted from further hybridization procedures.

**Statistical analysis**. Graphpad prism 6 was used to analyze data (Graphpad Software Inc.). Data is presented as mean + standard deviation (s.d.). To compare means of two groups, unpaired $t$-test was used and means of three or more groups were compared by ANOVA followed by Sidak's post-test. $p < 0.05$ was considered as statistically significant and was shown in figures as * for $p < 0.05$, ** for $p < 0.01$, *** for $p < 0.001$, **** for $p < 0.0001$. Gene expression statistical analysis was performed using Omics Explorer (Qlucore) and differentially expressed genes were determined with a $p$ value cutoff of 0.05 on Log2 transformed data.

**Reporting summary**. Further information on research design is available in the Nature Research Reporting Summary linked to this article.

## Data availability
All relevant data are available from the authors. Source data underlying Figs. 1e, 2d, 3f, 4f and Supplementary Figure 1b, 1d, 2a, 2b and 2d are provided as Source data file. The data discussed in this publication have been deposited in NCBI's Gene Expression Omnibus and are accessible through GEO Series accession number GSE132530, GSE132537, GSE123549.

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

## Acknowledgements

We thank Vuk Cerovic for helping with cell sorting, Ana Izcue and Vuk Cerovic for critical reading the manuscript and discussion, Anika Schridde for her help with cell isolation, Hildegard Ostendorp and Sabine von Oy for technical assistance, and Frank Tacke, Angela Schippers and Christoph Müller for providing transgenic mice. Cell sorting was supported by the Flow Cytometry Facility, a core facility of the Inter-disciplinary Center for Clinical Research (IZKF) Aachen within the Faculty of Medicine at RWTH Aachen University. G.D. received support from a scholarship from German Academic Exchange Service (DAAD), the center of infection biology (ZIB) from Hannover Medical School and Hannover Biomedical Research School (HBRS). This work was supported by German Research Foundation grant DFG 921/2–1, DFG 921/4–1 and CRC1382 B06 (to O.P.).

## Author contributions

G.D. performed the experiments and analyzed the data. O.P. conceived and designed the study. G.D. and O.P. wrote the manuscript.

## Additional information

**Competing interests:** The authors declare no competing interests.

