## [Peer Review File · Nature Communications]

Reviewers' comments:

Reviewer #1 (Mucosal immunology, macrophage)(Remarks to the Author):

In this manuscript, Desalegn & Pabst compare the processes which govern the recruitment and subsequent differentiation of monocytes in the intestinal mucosa under steady state and inflammatory conditions. As well as confirming previous evidence that CCR2 dependent accumulation of monocytes into the healthy mucosa is followed by progressive and tissue specific differentiation into resident macrophages, the current work presents the novel findings that the b7 integrin is important for recruitment of monocytes under steady state, but not inflamed conditions. In addition, the authors show that during inflammation induced by surgical manipulation, the recruited monocytes follow a distinct pathway of development to that found in healthy intestine, rather than undergoing a form of arrested development, as suggested previously. The work uses an elegant series of experimental approaches to make these conclusions, examining gene expression by donor monocytes found in tissues after transfer into recipients and the clearly presented data provide convincing evidence for the conclusions made. Given the small numbers of cells being examined, it is perhaps understandable that monocyte fate was not explored in more detail by studying a wider range of genes and/or functions. However this would have certainly increased the impact of the study, as would have some attempt to identify mechanisms involved in differentiation in different conditions. Specific comments:

- 1) As noted above, one limitation of the study is that the transcriptional analyses are based on a very small and highly selected sample of genes. While the resulting findings are interpreted appropriately, there must be a risk that similarities or differences between the various populations have been underestimated. For instance, are small intestinal and liver monocytes/macrophages really so alike as suggested by the current analysis? Are the differentiated monocytes found in healthy and inflamed mucosa on day 5 as similar as might appear here (in fact ~15-20% of the genes assessed seem to be differentially expressed). Examination of more genes may well also have revealed information of mechanistic interest and the authors may want to consider selective use of eg single cell RNAseq to substantiate and extend their findings.
- 2) The model of inflammation is elegant, but it would be useful to know how the endogenous populations of myeloid cells behave at different times after manipulation. This would allow clearer interpretation of how donor monocytes behave when transferred.
- 3) It would be important to show replicate data and statistical analysis for the phenotypic populations shown in Figures 3 and 4. Representative overlays comparing the expression of individual markers on specific subsets from healthy and inflamed intestine would also be useful.
- 4) Some of the heatmaps could be laid out more clearly. For instance, it would be helpful to have had the various genes shown in the same order in Figures 3 and 4, while it is not obvious why the replicate samples from the same tissue are not shown next to each other in Figures 5 and 6.
- 5) Figure 4g does not appear to show genes that "increased expression in steady state condition were not differentially expressed in inflamed small intestine"
- 6) Some analysis of bone marrow monocytes from inflamed mice would have been useful for confirming the authors' conclusions that the adoptive transfer of normal monocytes replicates the inflamed situation in vivo

Reviewer #2 (Transcriptome, macrophage)(Remarks to the Author):

In this manuscript, the authors utilize a mouse chimera model to study homing receptors influencing establishment of monocyte derived macrophages in the small intestine in steady state and following surgical manipulation. The most striking finding reported here is the sole requirement for Ccr2 expression on monocytes to enter the inflamed intestine, whereas in unmanipulated controls, expression of Itgb7 and Ccr2 on monocytes play important roles. The authors then perform

transcriptional profiling nanostring experiments and find early alterations in an inflammatory gene set in newly recruited cells during small intestine inflammation. The manuscript in its present form is carefully written and adequately supportive of the stated conclusions. However, enthusiasm for publication of this manuscript in Nature Communications is tempered by the lack of extension of the findings to biological significance (i.e., niche alterations supporting sole dependence on Ccr2 during inflammation, assessment of role for monocytes in promoting intestinal return to homeostasis, etc.). In addition, I have the following specific concerns:

Major:

1 "Recovery of adoptively transferred cells from the small intestine of WT recipients was lower as compared to CCR2-deficient recipients.... Thus for further adoptive transfer experiments we used CCR2-deficient mice as recipients unless stated otherwise." It is unclear recruited monocytes in the small intestine of CCR2 knockout mice reflect physiological monocyte recruitment and differentiation in WT mice. The authors could improve the manuscript by providing data supporting key findings using wild-type controls.

2 Small intestine inflammation controls were not sham, but instead received no surgical manipulation. This leads to questions as to whether the inflammatory signature in the newly recruited intestinal cells is solely due to the disease model, or partially due to surgical opening of the peritoneal cavity.

3 The authors show that monocytes recruited to the small intestine depend on CCR2 and β 7-integrin in the steady state, while the monocyte recruitment in inflamed small intestine doesn't depend on β 7-integrin. It is possible different subsets of circulating monocytes are recruited during intestinal inflammation. It is unclear whether the small intestine environment contributes to the difference in the transcriptional phenotype of recruited monocytes.

Minor :

1 In Fig. 3g and 4g, the authors describe the nanostring experiment as coming from pools of 2-3 independent experiments with a total of 4-5 mice for each time point. What does this actually mean because there are 4-5 columns per group not 2-3?

2 The use of Nanostring technology does support the main conclusions but could be missing a lot. The number of differential genes from the a priori list seems promising and could warrant redoing the experiment genome wide.

We would like to thank the reviewers for their constructive suggestions and helpful comments. Based on reviewer feedback we have now performed substantial additional experiments, included novel data and modified the text to reflect this. We believe that these changes address all points raised by the reviewers and, have resulted in a significantly improved manuscript. We will address the concerns raised by the reviewers in a point-by-point reply, outlined below.

Reviewers' comments:

Reviewer #1 (Mucosal immunology, macrophage)(Remarks to the Author):

In this manuscript, Desalegn & Pabst compare the processes which govern the recruitment and subsequent differentiation of monocytes in the intestinal mucosa under steady state and inflammatory conditions. As well as confirming previous evidence that CCR2 dependent accumulation of monocytes into the healthy mucosa is followed by progressive and tissue specific differentiation into resident macrophages, the current work presents the novel findings that the b7 integrin is important for recruitment of monocytes under steady state, but not inflamed conditions. In addition, the authors show that during inflammation induced by surgical manipulation, the recruited monocytes follow a distinct pathway of development to that found in healthy intestine, rather than undergoing a form of arrested development, as suggested previously. The work uses an elegant series of experimental approaches to make these conclusions, examining gene expression by donor monocytes found in tissues after transfer into recipients and the clearly presented data provide convincing evidence for the conclusions made. Given the small numbers of cells being examined, it is perhaps understandable that monocyte fate was not explored in more detail by studying a wider range of genes and/or functions. However this would have certainly increased the impact of the study, as would have some attempt to identify mechanisms involved in differentiation in different conditions.

We would like to thank the reviewer for the thoughtful review of our manuscript and highlighting the 'elegant' nature of our experimental approaches.

We have performed additional experiments and included new data in the revised manuscript. Most notably, we included new panels in Figure 3 and 4 and added the new Supplementary Figures 1, 3, 4, 5 and 6. At this point we have to admit that we had not been able to perform single cell RNA sequencing on adoptively transferred cells (see our response and arguments to this reviewer's first specific comment). This reviewer's specific comments 2, 3, 4, 5, and 6 have been fully addressed and are reflected by substantial changes in the manuscript and inclusion of new experimental data.

Additionally, we included a new experimental data set to study the role of Trem1 in directing macrophage differentiation in the small intestine. We report changes in Trem1 expression as monocytes differentiate into macrophages at different time points as well as differential expression of Trem1 between inflamed and non-inflamed conditions. To study the role of Trem1 in monocyte to macrophage differentiation in the gut, we used the adoptive monocyte transfer system comparing the transcriptomic changes in wild type monocytes to Trem1-deficient monocytes. We include this data to both demonstrate the viability of our experimental approach for identification of novel factors involved in monocyte to macrophage differentiation, but also to highlight the hitherto unappreciated role of Trem1 in this process.

Specific comments:

1) As noted above, one limitation of the study is that the transcriptional analyses are based on a very small and highly selected sample of genes. While the resulting findings are interpreted appropriately, there must be a risk that similarities or differences between the various populations have been underestimated. For instance, are small intestinal and liver monocytes/macrophages really so alike as suggested by the current analysis? Are the differentiated monocytes found in healthy and inflamed mucosa on day 5 as similar as might appear here (in fact ~15-20% of the genes assessed seem to be differentially expressed). Examination of more genes may well also have revealed information of mechanistic interest and the authors may want to consider selective use of eg single cell RNAseq to substantiate and extend their findings.

We are aware of the power of single cell sequencing to deduce developmental processes and have carefully considered this technique before starting this study and when preparing for this revision. Our manuscript builds on others' and our own previous work to study intestinal macrophages. Most importantly in previous work we used full transcriptome analysis on sort purified subpopulations of endogenous intestinal MNPs to study intestinal macrophage differentiation (Schridde at al., Mucosal Immunology 2017, ref 34 in the manuscript). Our transcriptome analysis revealed a progressive change in the transcriptomic profile of Ly6C⁺MHCII⁺ monocytes through an intermediary stage of Ly6C⁺MHCII⁺ to Ly6C⁻MHC^{hi} macrophages. However, the analysis of endogenous populations did not allow dissecting the precise kinetics of this process and did not differentiate between newly recruited cells and differentiation of cell that had been in the tissue.

Thus, here we aimed at solving these confounding issues and decided to use the adoptive monocyte transfer system as a complementary approach. We apologize but we have not been able to make this system work in combination with full transcriptome analysis and we are skeptical about the suitability of single cell sequencing in this set-up.

A typical single cell sequencing experiment will reveal only a low percentage of the actual transcriptome of a given cell. Thus besides 'sequencing depth' per cell a sufficient number of cells needs to be analyzed to obtain meaningful results in single cell sequencing experiments. Unfortunately, this is (virtually) impossible when performing adoptive monocyte transfer experiments. In non-manipulated recipients the number of adoptively transferred cells recovered from the recipient intestine is very low; in non-manipulated recipients at most only a few hundred cells can be detected. Considering the additional loss of cells when performing precise sorting, we decided to use 100 cells per sample (because this number of cells could be reliably sorted in all conditions to be analyzed including non-manipulated recipients and 'late' time points). With these numbers we cannot perform single cell sequencing and still cover a sufficient fraction of the whole transcriptome.

Thus, as strategic decision we established the nanostring analysis on a candidate gene set. The choice of genes included in the analysis is mostly based on our previous full transcriptome analysis of endogenous intestinal MNP and in fact with such a reduced list of genes one can fully recapitulate the transcriptomic changes that can be observed in subset of endogenous MNP in the colon of non-manipulated mice.

We agree with the reviewer that transcriptomic analysis of a broader set of genes would be attractive. Nonetheless we suggest that our approach outlined in this manuscript still provides exciting new information that is complementary to other work using full transcriptomic data on endogenous cell populations.

2) The model of inflammation is elegant, but it would be useful to know how the endogenous populations of myeloid cells behave at different times after manipulation. This would allow clearer interpretation of how donor monocytes behave when transferred.

We thank the reviewer for this comment and agree that this information helps the interpretation of the adoptive transfer experiments. We included a new Supplementary Figure 3 depicting endogenous populations of MNP in controls (CX3CR1gfp/+ mice) and compare these findings to CCR2-deficient mice (see also the first major comment by the second reviewer). Changes in phenotype of endogenous MNPs reflects the phenotypes observed in the adoptively transferred cells.

3) It would be important to show replicate data and statistical analysis for the phenotypic populations shown in Figures 3 and 4. Representative overlays comparing the expression of individual markers on specific subsets from healthy and inflamed intestine would also be useful.

We apologise for this omission. We have included new panels in Figure 3 and 4 (panels 3f and 4f) to depict frequencies of the different subsets. The revised panels 3g and 4g (panels 3f and 4f in the first submission) were modified to show overlays on specific subsets as suggested by the reviewer.

4) Some of the heatmaps could be laid out more clearly. For instance, it would be helpful to have had the various genes shown in the same order in Figures 3 and 4, while it is not obvious why the replicate samples from the same tissue are not shown next to each other in Figures 5 and 6.

We appreciate the important comments on the layout of the heatmap. The ordering of the genes is the results of the clustering algorithms and changing the ordering reduces the scientific information in the panels. We apologise for not making this clear in the original manuscript. The genes included in Figure 3h and 4h are limited to genes that are differentially expressed with a statistical significance p value of 0.05. To further improve readability of these data sets, we included a new Supplementary Figure 5 (panel 5b) listing all genes irrespective of their differential expression. To link this panel to Figure 3 and 4, gene names have been color-coded. Replicate samples in Figure 5 are shown 'next to each other' because the clustering algorithm has 'placed' the samples in this way. This grouped clustering reflects the high reproducibility of the replicate experiments. Indeed in Figure 6 replicate samples are not placed next to each other in some cases. The placing of the samples (like the ordering of genes) is determined by the clustering algorithm. In figure 6, the samples from the liver and small intestine are separated from those of BM as spleen, reflecting the differences in monocyte gene expression. However, the gene expression in the BM and spleen is similar enough that the replicate samples appeared intermingled, reflecting the high similarity of the monocytes in these two tissues.

5) Figure 4g does not appear to show genes that "increased expression in steady state condition were not differentially expressed in inflamed small intestine"

The original figure 4g (now 4h) shows all the genes that are significantly altered during the first five days of monocyte development in the inflamed intestine. The sentence in the original text was simply meant to highlight the absence of some genes that are differentially expressed at days 1-5 in the steady state small intestine.

Indeed some genes that are differentially expressed at days 1, 2 and 5 after adoptive transfer in steady state are not differentially expressed in inflamed intestine and vice versa. As described above we included a new Supplementary Figure 5 (panel 5b) to compile a complete list of all genes analyzed in the nanostring analysis for inflamed and non-inflamed intestine.

6) Some analysis of bone marrow monocytes from inflamed mice would have been useful for confirming the authors' conclusions that the adoptive transfer of normal monocytes replicates the inflamed situation in vivo

We thank the reviewer for this excellent suggestion. We performed additional experiments to phenotypically and functionally compare monocytes isolated from bone marrow of non-manipulated donors and donors with intestinal inflammation (depicted in the new supplementary Figure 6). Monocytes isolated from mice with intestinal inflammation resembled monocytes isolated from non-manipulated mice. Moreover, after adoptive transfer into non-manipulated recipients, monocytes isolated from donors with established intestinal inflammations were similar in their surface phenotype and expression profile to donor monocytes isolated from non-manipulated donors. These data suggest that the intestinal environment in the recipients and not the nature of the donor monocytes is the main determinant of monocyte differentiation in the recipient.

We would like to thank the reviewers for their constructive suggestions and helpful comments. Based on reviewer feedback we have now performed substantial additional experiments, included novel data and modified the text to reflect this. We believe that these changes address all points raised by the reviewers and, have resulted in a significantly improved manuscript. We will address the concerns raised by the reviewers in a point-by-point reply, outlined below.

Reviewer #2 (Transcriptome, macrophage)(Remarks to the Author):

In this manuscript, the authors utilize a mouse chimera model to study homing receptors influencing establishment of monocyte derived macrophages in the small intestine in steady state and following surgical manipulation. The most striking finding reported here is the sole requirement for Ccr2 expression on monocytes to enter the inflamed intestine, whereas in unmanipulated controls, expression of Itgb7 and Ccr2 on monocytes play important roles. The authors then perform transcriptional profiling nanostring experiments and find early alterations in an inflammatory gene set in newly recruited cells during small intestine inflammation. The manuscript in its present form is carefully written and adequately supportive of the stated conclusions. However, enthusiasm for publication of this manuscript in Nature Communications is tempered by the lack of extension of the findings to biological significance (i.e., niche alterations supporting sole dependence on Ccr2 during inflammation, assessment of role for monocytes in promoting intestinal return to homeostasis, etc.). In addition, I have the following specific concerns:

Major:

1 “Recovery of adoptively transferred cells from the small intestine of WT recipients was lower as compared to CCR2-deficient recipients.... Thus for further adoptive transfer experiments we used CCR2-deficient mice as recipients unless stated otherwise.” It is unclear recruited monocytes in the small intestine of CCR2 knockout mice reflect physiological monocyte recruitment and differentiation in WT mice. The authors could improve the manuscript by providing data supporting key findings using wild-type controls.

We thank the reviewer for this point, and we agree that it raises an important concern. Indeed the number of adoptively cells that can be recovered from non-manipulated wild type mice is too low to allow for a robust analysis (Figure R1, for the review process only).

Figure R1: Monocytes were transferred into non-manipulated wild type recipients. Four days after transfer, cells were isolated from the recipient intestine and the analyzed by flow cytometry. FACS plots are gated on adoptively transferred cells identified by congenic marker and pre-gated on CD11b-expressing cells. Based on the too low recovery of adoptively transferred cells from wild type recipients we decided to explore CCR2-deficient mice as recipients. Indeed, an appreciably greater number of cells could be recovered from CCR2-deficient mice as compared to wild type recipients. Plots represent two individual mice analyzed four days after adoptive monocyte transfer.

To challenge our approach of using CCR2-deficient mice, we have systematically compared the phenotype of endogenous MNP in the gut of wild type and CCR2-deficient mice by flow cytometry and analysis of their transcriptomic fingerprint. Expectedly cell numbers greatly differ between CX3CR1 reporter mice (used to reflect the wild type situation here) and CCR2-deficient mice. Nonetheless, qualitatively the behavior of endogenous MNP in CCR2-deficient mice seems to faithfully recapitulate the phenotype of MNP in wild type mice. These data are displayed in the new Supplementary Figure 3. Our data suggest that it is justified to use CCR2-deficient mice as recipients to study the differentiation of wild type monocytes into macrophages.

2 Small intestine inflammation controls were not sham, but instead received no surgical manipulation. This leads to questions as to whether the inflammatory signature in the newly recruited intestinal cells is solely due to the disease model, or partially due to surgical opening of the peritoneal cavity.

This is an excellent comment, and something we should have made clear in the original text. The phenotype of SI monocytes is not affected by sham surgeries. We apologise for this omission and have now included a new data set to describe the composition of myeloid cells in the small intestine of non-manipulated mice, mice that underwent laparotomy only (sham) and mice that underwent laparotomy plus mechanical manipulation of the gut to induce inflammation (new Supplementary Figure 1). For animal welfare reasons and because we aimed at comparing inflammation to no-inflammation - our interest is not primarily to study the effects of intestinal manipulation - we use this approach as model to induce intestinal inflammation), we did not systematically use Sham-surgery in all experiments.

3 The authors show that monocytes recruited to the small intestine depend on CCR2 and β 7-integrin in the steady state, while the monocyte recruitment in inflamed small intestine doesn't depend on β 7-integrin. It is possible different subsets of circulating monocytes are recruited during intestinal inflammation. It is unclear whether the small intestine environment contributes to the difference in the transcriptional phenotype of recruited monocytes.

We agree with the reviewer that unappreciated heterogeneity among bone marrow monocytes could account for some of the differences in recruitment seen when using CCR2 or β 7-integrin-deficient donors. To explore this question we have performed a careful comparison of Ly6C⁺ monocytes for a panel of markers (New Supplementary Figure 4). For comparison we analysed Ly6C low monocytes and included blood in addition to bone marrow. Ly6C⁺ monocytes displayed as uniform population in both compartments and for all markers investigated..

Moreover, we compared the phenotype of monocytes isolated from bone marrow of non-manipulated mice and mice with established gut inflammation (after surgical gut manipulation) (depicted in the new supplementary Figure 6). Monocytes isolated from mice with intestinal inflammation resembled monocytes isolated from non-manipulated mice. Moreover, after adoptive transfer into non-manipulated recipients, monocytes isolated from donors with established intestinal inflammations were similar in their gene expression to donor monocytes isolated from non-manipulated donors. These data suggest that the intestinal environment in the recipients and not the nature of the donor monocytes is the main determinant of monocyte differentiation in the recipient.

Minor :

1 In Fig. 3g and 4g, the authors describe the nanostring experiment as coming from pools of 2-3 independent experiments with a total of 4-5 mice for each time point. What does this actually mean because there are 4-5 columns per group not 2-3?

We apologize for this inaccuracy. All samples/all columns in Figure 3h and 4h (Panel 3g and 4g in the first version of the manuscript) show the analysis of individual mice. Mice have been pooled from several (2-3 independent) experiments. Considering the effort required to obtain a sufficient number of donor monocytes and even more the time consuming sort, in each experiment 1-2 recipient/mice were included/analyzed per group.

Independent experiments have been performed on different days and included independent monocyte isolation, transfer, housing of the mice and sort purification. We amended the manuscript to more clearly describe the nature of our data.

2 The use of Nanostring technology does support the main conclusions but could be missing a lot. The number of differential genes from the a priori list seems promising and could warrant redoing the experiment genome wide.

The Nanostring approach in combination with candidate genes is indeed not the method of choice to identify new candidate genes to be analyzed for the function in monocyte to macrophage differentiation. We have previously used array analysis to identify genes that might potentially direct macrophage differentiation in a gut-specific manner and numerous other group have used such approaches to identify tissue specific signatures of macrophages. In fact the list of candidate genes used here is mostly based on such analysis.

To further substantiate the technical approach followed in this study, we included new experimental data to study the role of Trem1 in directing macrophage differentiation in the small intestine. Our observations in wild type mice revealed changes in Trem1 expression as monocytes differentiate into macrophages at different time points as well as differential expression of Trem1 between inflamed and non-inflamed conditions (depicted in the new Supplementary Figure 5). To study the role of Trem1 in monocyte to macrophage differentiation in the gut, we used the adoptive monocyte transfer system comparing the transcriptomic changes in wild type monocytes to Trem1-deficient monocytes. We include this data to both demonstrate the viability of our experimental approach for identification of novel factors involved in monocyte to macrophage differentiation, but also to highlight the hitherto unappreciated role of Trem1 in this process

Reviewers' comments:

Reviewer #1 (Remarks to the Author):

I thank the authors for their thoughtful and detailed responses to my original comments. The additional data and revised presentation are extremely helpful and raise the impact of the study significantly. I am not entirely sure how the experiment using Trem1^{-/-} monocytes adds to the work, especially as no unmanipulated recipients have been used as a control, but the data are novel and will probably be of interest to the community.

Reviewer #2 (Remarks to the Author):

In their revision, the authors adequately address many comments raised by both reviewers. However, in the view of this reviewer, extension of the findings to provide biological significance remains limited. The author's major effort at addressing this comment was inclusion of additional nanostring experiments assessing the role of Trem1 in coordinating "normal" monocyte to macrophage differentiation during inflammation. The authors did not extend transcriptional studies to an unbiased comparison of peripheral monocytes nor intestinal recruited monocytes from Trem1 knockouts to control. While this is not essential, the authors did not show that Trem1 was involved in a pathophysiological consequence associated with the transcriptomic change.

Additional concerns:

1. In figure R1, they are showing the number of adoptively transferred cells that can be recovered from non-manipulated wild type mice. However, it is more important to show the data in manipulated wild type mice. They also need to show the character of recruited monocytes are similar in wild type and Ccr2 knockout mice in inflamed condition.
2. In lines 208-209, they describe "To refine the characterization of monocyte differentiation, we identified a set of 67 genes (Supplementary Table 1) to reflect different developmental stages and functional differences between tolerogenic and pro-inflammatory monocytes/macrophages 11, 12, 34. Genes in our study were carefully selected to recapitulate main transcriptomic differences of the known stages of monocyte to macrophage differentiation in the non-inflamed colon 34". Since organ-specific factors are not considered, it is inadequate to apply this gene set to evaluate the gene expression profiles of recruited monocytes in other organs (PMID: 23023392; PMID: 27492475; PMID: 25480296; PMID: 25480297)
3. Similarly, it is not appropriate to apply this gene set to examine differential expressed genes in wild type and Trem1 knockout monocytes from bone marrow. Even if only 2/72 genes are differentially expressed in Trem1 knockout cells, there may be other critical gene expression changes before the monocytes transfer. They also should show the gene expression profile of Trem1 knockout monocytes in non-inflamed condition.

We would like to thank the reviewers for their constructive suggestions and helpful comments. Based on reviewer feedback, we included additional experimental results. In particular we performed adoptive transfer of Trem1-deficient monocytes into non-inflamed recipients and included a comparison of adoptively transferred monocytes recovered from manipulated wild type and CCR2-deficient recipients in our revised manuscript. We will address the concerns raised by the reviewers in a point-by-point reply, outlined below. Changes in the manuscript are marked by underlines.

Reviewers' comments:

Reviewer #1 (Remarks to the Author):

I thank the authors for their thoughtful and detailed responses to my original comments. The additional data and revised presentation are extremely helpful and raise the impact of the study significantly. I am not entirely sure how the experiment using Trem1^{-/-} monocytes adds to the work, especially as no unmanipulated recipients have been used as a control, but the data are novel and will probably be of interest to the community.

We are grateful for the positive feedback on our revised manuscript. To further enhance the dataset, we have included new experimental results demonstrating the phenotype of Trem1-deficient monocytes transferred into non-manipulated (non-inflamed) recipients. These data are displayed in the revised Supplementary Figure 6 (part of supplementary Figure 5 in the previous version of the manuscript).

Reviewer #2 (Remarks to the Author):

In their revision, the authors adequately address many comments raised by both reviewers. However, in the view of this reviewer, extension of the findings to provide biological significance remains limited. The author's major effort at addressing this comment was inclusion of additional nanostring experiments assessing the role of Trem1 in coordinating "normal" monocyte to macrophage differentiation during inflammation. The authors did not extend transcriptional studies to an unbiased comparison of peripheral monocytes nor intestinal recruited monocytes from Trem1 knockouts to control. While this is not essential, the authors did not show that Trem1 was involved in a pathophysiological consequence associated with the transcriptomic change.

Indeed our data do not establish a function of Trem1 in intestinal monocytes. Considering the far reaching effects of Trem1 on the expression profile of intestinal monocytes/macrophages this may become an exciting topic for further research. As suggested by the reviewer, we added new experimental data to demonstrate the effects of Trem1 deficiency on monocytes adoptively transferred into non-manipulated/non-inflamed recipients. These data are displayed in the revised Supplementary Figure 6 (part of supplementary Figure 5 in the previous version of the manuscript).

Additional concerns:

1. In figure R1, they are showing the number of adoptively transferred cells that can be recovered from non-manipulated wild type mice. However, it is more important to show the

data in manipulated wild type mice. They also need to show the character of recruited monocytes are similar in wild type and Ccr2 knockout mice in inflamed condition.

We apologize, we included Figure R1 in our previous point-by-point reply only to illustrate the very low number of adoptively transferred cell recovered from non-manipulated wild type recipients. The cell yield in non-manipulated wild type recipients does not allow for a robust analysis and cell sorting.

This situation is different in manipulated wild type recipients. Manipulation/intestinal inflammation results in enhanced recruitment of monocytes in wild type as well as in CCR2 deficient recipients. We included a new data set comparing the phenotype of adoptively transferred MNP^{CD64+} cells (the recruited cells) in manipulated CCR2-deficient recipients and manipulated wild type recipients. These data are show in the new Supplementary Figure 5b.

Figure R2 (identical to the new/revised Supplementary Fig. 5b): Representative FACS plots comparing phenotypic changes of donor MNP^{CD64+} cells over time, with respect to their expression of Ly6C and MHCII. The upper panel demonstrates the phenotype of MNP^{CD64+} cells in CCR2 deficient recipients, the lower panel shows the phenotype in wild type recipients.

2. In lines 208-209, they describe “To refine the characterization of monocyte differentiation, we identified a set of 67 genes (Supplementary Table 1) to reflect different developmental stages and functional differences between tolerogenic and pro-inflammatory monocytes/macrophages 11, 12, 34. Genes in our study were carefully selected to recapitulate main transcriptomic differences of the known stages of monocyte to macrophage differentiation in the non-inflamed colon 34”. Since organ-specific factors are not considered, it is inadequate to apply this gene set to evaluate the gene expression profiles of recruited monocytes in other organs (PMID: 23023392; PMID: 27492475; PMID: 25480296; PMID: 25480297)

As discussed in our previous response, we agree that the nanostring analysis may miss important differences in gene regulation/changes in gene expression pattern. As compared to the gut, this problem may be more relevant in organs that have not been considered when selecting the gene list for nanostring analysis. We added additional text to the manuscript to discuss this potential problem.

3. Similarly, it is not appropriate to apply this gene set to examine differential expressed genes in wild type and Trem1 knockout monocytes from bone marrow. Even if only 2/72 genes are

differentially expressed in Trem1 knockout cells, there may be other critical gene expression changes before the monocytes transfer.

To avoid a potential overstatement, we removed the panel demonstrating the analysis of bone marrow monocytes. The finding is described in the text as follows: 'The nanostring analysis revealed no major differences between Trem1-deficient and WT bone marrow monocytes (out of the 67 genes analyzed only two genes, IL23a and CXCL9, showed significant differences in their expression). This suggests that Trem1 did not have a major effect on the phenotype of bone marrow monocytes with respect to these genes (but does not exclude potential effects of Trem1 on the overall monocyte transcriptome).'

They also should show the gene expression profile of Trem1 knockout monocytes in non-inflamed condition.

As suggested by the reviewer we added new experimental data to demonstrate the effects of Trem1 deficiency on monocyte adoptively transferred into non-manipulated/non-inflamed recipients. These data are displayed in the revised Supplementary Figure 6.

REVIEWERS' COMMENTS:

Reviewer #2 (Remarks to the Author):

Although the authors do not perform global RNA seq experiments, the conclusions are adequately supported and the revisions/clarifications satisfy my major remaining concerns.